

# Sensitivity of Uncertainty in Wind Characteristics and Wind Turbine Properties on Wind Turbine Extreme and Fatigue Loads

Amy N. Robertson[1], Kelsey Shaler[1], Latha Sethuraman[1], Jason Jonkman[1]

[1]National Renewable Energy Laboratory, 15013 Denver West Parkway, Golden, CO 80401, USA

*Correspondence to*: Amy N. Robertson (amy.robertson@nrel.gov)

**Abstract.** Wind turbine design relies on the ability to accurately predict turbine ultimate and fatigue loads. The loads analysis process requires precise knowledge of the expected wind-inflow conditions as well as turbine structural and aerodynamic properties. However, uncertainty in most parameters is inevitable. It is therefore important to understand the impact such uncertainties have on the resulting loads. The goal of this work is to assess which input parameters have the greatest influence

on turbine power, fatigue loads, and ultimate loads during normal turbine operation. An Elementary Effects (EE) sensitivity analysis is performed to identify the most sensitive parameters. Separate case studies are performed on 1) wind-inflow conditions and 2) turbine structural and aerodynamic properties, both using the National Renewable Energy Laboratory (NREL) 5-MW baseline wind turbine. The focus is on individual parameter sensitivity, though interactions between parameters are considered.

The results of this work show that for wind-inflow conditions, turbulence in the primary wind direction and shear are the most sensitivity parameters for turbine loads, which is expected. Secondary parameters of importance are identified as veer, u-direction integral length, and components of the IEC coherence model ($a_u$ and $b_u$), as well as the exponent ($\gamma$). For the turbine properties, the most sensitive parameters are yaw misalignment ($\theta$) and outboard lift coefficient ($C_{l,t}$) distribution. This

information can be used to help establish error bars around the predictions of engineering models during validation efforts, and provide insight to probabilistic design methods and site-suitability analyses.

## 1 Introduction

Wind turbines are designed using the IEC 61400-1 standard, which prescribes a set of simulations to ascertain the ultimate and fatigue loads that the turbine could encounter under a variety of environmental and operational conditions. The standard applies

safety margins to account for the uncertainty in the process, which comes from the procedure used to calculate the loads (involving only a small fraction of the entire lifetime), but also from uncertainty in the properties of the system, variations in the conditions the turbine will encounter from the prescribed values, and modeling uncertainty. As manufacturers move to develop more advanced wind technology, better optimized designs, and reduce the cost of wind turbines, it is important to better understand how uncertainties impact modeling predictions and reduce the uncertainties where possible. Knowledge of

where the uncertainties stem from can lead to a better understanding of the cost impacts and design needs of different sites and different turbines.

This paper provides a better understanding of the uncertainty in the ultimate and extreme structural loads and power in a wind turbine. This is done by parameterizing the uncertainty sources; prescribing a procedure to estimate load sensitivity to each

parameter; and identifying which parameters have the largest sensitivities for a conventional utility-scale wind turbine under normal operation. This is a first step in understanding potential design process modifications to move towards a more probabilistic approach or to inform site-suitability analyses. The results of this work can be used to 1) rank the sensitivities of different parameters, 2) help establish error bars around the predictions of engineering models during validation efforts, and 3) provide insight to probabilistic design methods.





## 2 Analysis Approach

### 2.1 Overview

To identify the most influential sources of uncertainty in the calculation of the structural loads for utility-scale wind turbines, a sensitivity analysis methodology based on Elementary Effects (EE) is employed. The focus is on the sensitivity of the input
parameters of wind turbine simulations (used to calculate the loads), not on the modeling approach itself, which creates uncertainty based on whether the approach used accurately represents the physics of the wind loading and structural response. The procedure followed is summarized in the following sub-sections. The caveats of the sensitivity analysis approach employed are given as follows:

- Only the NREL 5-MW turbine is used to assess sensitivity (the resulting identification of most-sensitive parameters
may depend on the turbine).
- Only normal operation under turbulence is considered (gusts, start-ups, shut-downs, and parked/idling events are not considered).
- Min/max values of the input parameter uncertainty ranges are examined in the analysis (no joint probability density function is considered).
- With the exception of wind speed, each parameter is examined independently across the full range of variation, and is not conditioned based on other parameters.

### 2.2 Wind turbine model and tools

The sensitivity of each input parameter on turbine load response is assessed through the use of a simulation model. The NREL 5-MW reference turbine (Jonkman et al., 2009) was used in this study as a representative turbine. This is a variable-speed, 3-
bladed, upwind, horizontal-axis turbine with a hub height of 90 m and a rotor diameter of 126 m. Though not covered in this work, it would also be useful to examine in future work how the sensitivity of the parameters on turbine loads is affected by the size and type of the considered wind turbine.

The effect of input parameters on load sensitivity could be influenced by the wind speed and associated wind turbine controller
response. Therefore, the EE analysis was performed at three different wind speeds corresponding to mean hub height wind speeds of 8, 12, and 18 m/s, representing below-, near-, and above-rated wind speeds, respectively. Turbulent wind conditions were generated at each wind speed using TurbSim (Jonkman, 2009), employing an IEC Kaimal turbulence spectrum with exponential spatial coherence. Multiple turbulence seeds were used for each input parameter variation to ensure the variation from input parameter changes is distinguishable from the variation of the selected turbulence seed. The number of seeds was
determined through a convergence study for each of the parameter sets. A 25 x 25 point square grid of three-component wind vector points that encompasses the turbine rotor plane was used.

OpenFAST, a state-of-the-art engineering-level aero-servo-elastic modeling approach, was used to simulate the NREL 5-MW wind turbine using the developed wind files, allowing for aeroelastic response and turbine operation analysis. A simulation
time of 10 minutes was used after an initial 30-second transient period per turbulence seed. Drag on the tower was not considered because it is negligible for an operational turbine. AeroDyn, the aerodynamic module of OpenFAST, determines the impact of the turbine wake using induction factors that are computed using blade-element momentum (BEM) theory with advanced corrections. Steady and unsteady aerodynamic response were considered. Steady aerodynamic modeling uses static lift and drag curves in the momentum balance to calculate the local induction. Unsteady airfoil aerodynamic modeling accounts
for dynamic stall, flow separation, and flow reattachment to calculate the local aerodynamic applied loads. ElastoDyn, a combined multibody and modal structural approach that includes geometric nonlinearities, was used to represent the flexibility of the blades, drivetrain, and tower and compute structural loading, which was used to compute ultimate and fatigue loads. The baseline controller of the NREL 5-MW turbine was enabled using ServoDyn. OpenFAST results were used to assess the change in response quantities of interest (QoIs) to changes in the physical input parameters.





### 2.3 Case Studies

Input parameters were identified that could significantly influence the loading of a utility-scale wind turbine. These parameters were organized into two main categories (or case studies): the ambient wind-inflow conditions that will generate the aerodynamic loading on the wind turbine and the aeroelastic properties of the structure that will determine how the wind

turbine will react to that loading (see Figure 1). Within these two categories, a vast number of uncertainty sources can be identified, and Abdallah (2018) provides an exhaustive list of the properties. For this study, the authors down-selected those parameters believed to have the largest effect for normal operation for a conventional utility-scale wind turbine, which are categorized into the labels shown in Figure 1.

To understand the sensitivity of a given parameter, a range over which that parameter may vary needed to be defined. For the wind conditions, a literature search was done to identify the reported range for each of the parameters across different potential wind-farm locations within the three wind speed bins. For the aeroelastic properties, the parameters are varied based on an assessed level of potential uncertainty associated with each parameter.

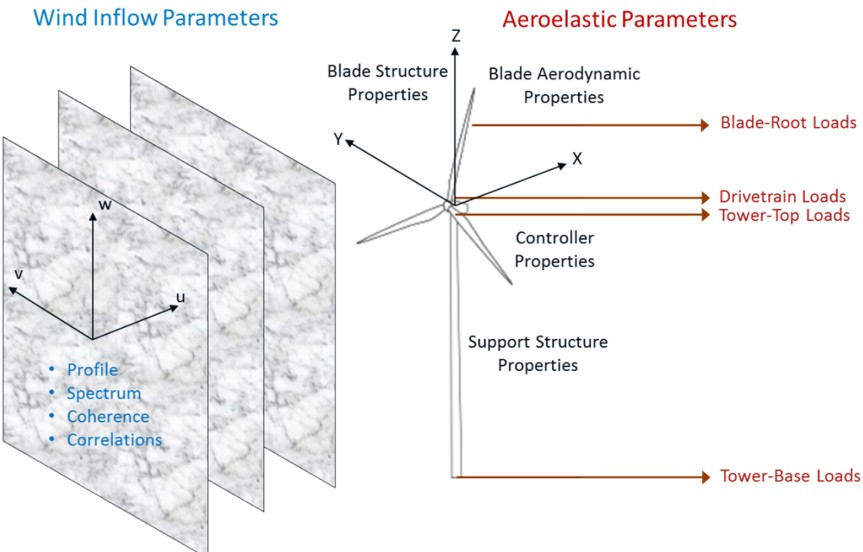

**Figure 1: Overview of the parametric uncertainty in a wind turbine loads analysis. Includes wind-inflow conditions (subset shown in blue), turbine aeroelastic properties (subset shown in black), and the associated load QoI (subset shown in red).**

### 2.4 Quantities of Interest

To capture the variability of turbine response that results from parameter variation, several QoIs were identified. These QoIs are summarized in Table 1 and include the blade, drivetrain, and tower loads; blade-tip displacement; and turbine power.

Ultimate and fatigue loads were considered for all load QoIs, whereas only ultimate values were considered for blade-tip displacements. The ultimate loads were estimated using the average of the global absolute maximums across all turbulence seeds for a given set of parameter values. The fatigue loads were estimated using damage-equivalent loads (DEL) of the output response across all seeds for a given set of parameter values. For the bending moments, the ultimate loads were calculated as the largest vector sum of the first two components listed, rather than considering each individually. The QoI sensitivity of each

parameter is examined using the procedure summarized in the next section.



Table 1. Quantities of interest examined in the sensitivity analyses.

| Quantity of Interest | Component | | |
|---|---|---|---|
| Blade-root moments | Out-of-plane bending | In-plane bending | Pitching moment |
| Low-speed shaft moments at main bearing | 0-degree bending | 90-degree bending | Shaft torque |
| Tower-top moment | Fore/aft bending | Side/side bending | Yaw moment |
| Tower-base moment | Fore/aft bending | Side/side bending | |
| Blade-tip displacement | Out-of-plane (Ultimate only) | | |
| Electrical power | | | |

## 3 Sensitivity Analysis Procedure

### 3.1 Sensitivity Analysis Approaches

There are many different approaches to assess the sensitivity of the QoI for a given input parameter. The best choice depends on the number of considered input parameters, simulation computation time, and availability of parameter distributions. Sensitivity is commonly assessed through the Sobol sensitivity (Saltelli et al., 2008), which decomposes the variance of the response into fractions that can be attributed to different input parameters and parameter interactions. The drawback of this method is the large computational expense, which requires a Monte Carlo analysis to calculate the sensitivity. To decrease the computational expense, one approach is to use a meta-model, which is a lower-order surrogate model trained on a subset of
simulations to capture the trends of the full-order, more computationally expensive model. This approach has been used in the wind energy field (Nelson et al., 2003; Rinker, 2016; Sutherland, 2002; Ziegler et al., 2016), but was deemed unsuitable for this work given the wind turbine model complexity and associated QoIs. Specifically, it may be difficult for a meta-model to capture the system nonlinearities and interaction of the controller, especially the ultimate loads, limiting meta-model accuracy. Another approach to reduce computational expense is to use a design of experiments approach to identify the fewest
simulations needed to capture the variance of the parameters and associated interactions, *e.g.*, Latin hypercube sampling (Matthaus et al., 2017; Saranyasoontorn, 2006; Saranyasoontorn et al., 2008) and fractional factorial analysis (Downey, 2006). These methods were considered for this application, but such approaches are still too computationally expensive given the large number of considered input parameters. Instead, a screening approach was determined to be the best approach. A screening method provides a sensitivity measure that is not a direct estimate of the variance, but rather supplies a ranking of
those parameters with the most influence. One of the most commonly used screening approaches is called EE analysis (Compolongo et al., 2007; Campolongo et al., 2011; Francos et al., 2003; Gan, 2014; Huang et al., 2012; Jansen, 1999; Martin et al., 2016; Saint-Geours et al., 2010; Soheir et al., 2015). Once the EE analysis identifies the input parameters that are most influential to the QoIs, a more targeted analysis can be performed using one of the other sensitivity analyses discussed above.

### 3.2 Overview of Elementary Effects

EE at its core is a simple methodology for screening parameters. It is based on the one-at-a-time approach in which each input parameter of interest is varied individually while holding all other parameters fixed. A derivative is then calculated based on the level of change in the QoI to the change in the input parameter using first-order finite differencing. Approaches such as these are called local sensitivity approaches because they calculate the influence of a single parameter without considering interaction with other parameters. However, the EE method extends this process by examining the change in response for a
given input parameter at different locations (points) in the input parameter hyperspace. In other words, only one parameter is varied at a time, but this variation is performed multiple times using different values for the other input parameters, as shown in Figure 2. The derivatives calculated from the different points are considered to assess an overall level of sensitivity. Thus, the EE method considers the interactions between different parameters and is therefore considered a global sensitivity analysis method.



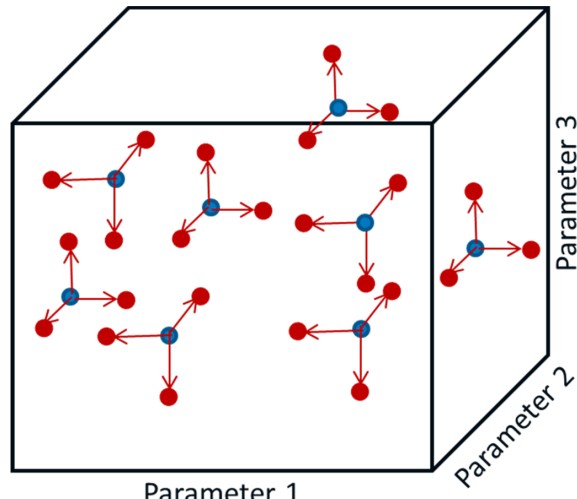

**Figure 2: Radial EE approach representation for 3 input parameters. Blue circles indicate starting points in the parameter hyperspace. Red points indicate variation of one parameter at a time. Each variation is performed for 10% of the range over which the parameter may vary, either in the positive or negative direction.**

Each wind turbine QoI, $Y$, is represented as a function of different characteristics of the wind or model property input parameters, $U$, as follows:

$$Y = f(U_1, U_2, U_3, ... U_I) \qquad (1)$$

where $I$ is the total number of input parameters. In the general EE approach, all input parameters are normalized between 0 (minimum value) and 1 (maximum value). For a given sampling of $U$, the EE value of the $i^{th}$ input parameter is found by
varying only that parameter by a normalized amount, $\Delta$:

$$EE_i = \frac{Y(U_1,...,U_{i-1}, x_i + \Delta, U_{i+1},...,U_I) - Y(U_1, U_2,...,U_I)}{\Delta} \qquad (2)$$

Because of the normalization of $U$, the elementary effect ($EE_i$) value can be thought of as the local partial derivative of the output QoI ($Y$) with respect to an input ($U_i$), scaled by the range of that input. Thus, the EE value has the same unit as the
output QoI, $Y$. The EE value is calculated for $R$ starting points in the input parameter hyperspace, creating a set of $R$ different calculations of EE value for each input parameter.
The basic approach for performing an EE analysis has been modified over the years to ensure that the input hyperspace is being adequately sampled and to eliminate issues that might confound the sensitivity assessment. In this work, the following modifications to the standard approach were made:

1.  A radial approach was used rather than the traditional trajectories for varying all of the parameters, which has been shown to improve the efficiency of the method.
2.  Sobol numbers were used to determine the initial points at which the derivatives will be calculated (blue circles in Figure 2), which ensures a wide sampling of the input hyperspace (Campolongo et al., 2011; Robertson et al., 2018).
3.  A set delta value equal to 10% of the input parameter range was used to ensure the calculation of the finite difference
occurred over an appropriate range to better meet the assumption of linearity.




4. A modified EE formula—different for ultimate and fatigue loads—was used to examine the sensitivity of the parameters across multiple wind speed bins.

**3.3 Elementary Effects Formulas**

This section provides the detailed formulas used to calculate the EE values for the ultimate and fatigue loads.

**3.3.1 Ultimate Loads**

When considering the ultimate loads, only the single highest ultimate load is of concern, regardless of the wind speed bin. Therefore, the standard EE formula is modified so that the sensitivity of the parameters can be examined consistently across different wind speed bins. This is accomplished by keeping $U$ and $\Delta$ dimensional (i.e., not nondimensionalizing $U$ between 0 and 1), multiplying the derivative by the total range of the input for a given wind speed bin, and adding the nominal value of

the QoI associated with IEC turbine class $I$ and category $B$ for the given wind speed bin. The EE of input parameter $U_{ib}^r$ on a certain QoI (output), $o$, for starting point $r$ in wind speed bin $b$ is then given by:

$$EE_{oib}^r = \left| \frac{Y_o(U_{1b}^r, \dots, U_{ib-1}^r, U_{ib}^r + \Delta_{ib}, U_{ib+1}^r, \dots U_{lb}^r) - Y_o(U_{1b}^r, U_{2b}^r, \dots \dots U_{lb}^r)}{\Delta_{ib}} (U_{ibmax} - U_{ibmin}) \right| + \bar{Y}_{ob} \qquad (3)$$

where the ultimate load, $Y_o(\ )$, is defined as the mean of the absolute maximum of the temporal response load in bin $b$ across

$S$ seeds for a certain input parameter $i$ and starting point $r$:

$$Y_o(\ ) = \frac{1}{S} \sum_{s=1}^{S} MAX(|Y_{os}(\ )|) \qquad (4)$$

$MAX(|Y_{os}(\ )|)$ is the absolute maximum of a temporal response load in bin $b$ for starting point $r$ and seed $s$. Additionally, $\Delta_{ib}$ $=+/- \frac{(U_{ibmax} - U_{ibmin})}{10}$ where $U_{ibmin}$ and $U_{ibmax}$ are the lower and upper bounds for an input parameter $i$ in bin $b$ (the + or – sign

is chosen randomly). The added term ($\bar{Y}_{ob}$) represents the IEC-Class IB nominal value for the given wind speed bin.

**3.3.2 Fatigue Loads**

To compute the fatigue loads, the same basic formulation is used as for the ultimate loads, but the DEL of the temporal response is considered in place of the mean of the absolute maximums:

$$EE_{oib}^r = P(v_b) \left| \frac{DEL_o^{STF}(U_{1b}^r, \dots, U_{ib-1}^r, U_{ib}^r + \Delta_{ib}, U_{ib+1}^r, \dots U_{lb}^r) - DEL_o^{STF}(U_{1b}^r, U_{2b}^r, \dots \dots U_{lb}^r)}{\Delta_{ib}} (U_{ibmax} - U_{ibmin}) \right| \qquad (5)$$

where $DEL_o^{STF}(\ )$ is the aggregate of the short-term DEL of output $o$ (QoI) across all $S$ seeds computed using the NREL post-processing tool, MLife (Hayman et al., 2012). DELs are computed without the Goodman correction and with load ranges about a zero fixed mean. The fatigue EE value is scaled by $P(v_b)$, which is the Rayleigh probability at the wind speed $v_b$ (assuming

IEC-Class IB prescription) associated with bin $b$ to compare the fatigue loads consistently across wind speed bins.





### 3.4 Identification of Most Sensitive Inputs

The EE value is a surrogate for a sensitivity level. Therefore, a higher EE value for a given input parameter indicates more sensitivity. Here, the most sensitive parameters are identified by defining a threshold over which an individual EE value would be considered significant, indicating the sensitivity of the associated parameter. This approach differs from the classical method
of determining parameter sensitivity, as discussed in Appendix A. The threshold is set individually for each QoI. For the wind parameter study, the threshold is defined as $\overline{EE_{oib}^r} + 2\sigma$, where $\overline{EE_{oib}^r}$ is the mean of all EE values across all starting points $R$, inputs $I$, and wind speed bins $B$ for QoI $o$, and $\sigma$ is the standard deviation of these EE values. For the turbine parameter study, the threshold is modified to $\overline{EE_{oib}^r} + 1.7\sigma$ due to the stratification of the results based on wind speed bin. Additionally, the ultimate load thresholds for the turbine parameter study are computed using only near- and above-rated results because of the
separation of EE values between the below-, near-, and above-rated wind speed bins. However, below-rated results are included when the associated EE values occasionally exceed the threshold. For both studies, fatigue load EE values are not clearly separated by wind speed; therefore, all wind speeds are used to compute the fatigue load parameter thresholds.

### 4 Results

Two separate case studies were performed to assess the sensitivity of input parameters on the resulting ultimate and fatigue
loads of the NREL 5-MW wind turbine. The categories of input parameters analyzed were the wind-inflow conditions and the aeroelastic turbine properties. In both of the case studies, loads were analyzed for three wind speed bins, using mean wind speed bins of 8, 12, and 18 m/s, representing below-, near-, and above-rated wind speeds, respectively. Turbulent wind conditions were generated using an IEC Kaimal turbulence spectra with exponential spatial coherence functions. For the turbine parameter study, turbulence was based on IEC-Class IB turbulence.

### 4.1 Wind-Inflow Characteristics

Many researchers have examined the influence of wind characteristics on turbine load response, considering differing wind parameters and turbulence models, and using different methods to assess their sensitivity. The most common parameter considered is the influence of turbulence intensity variability, which past work has shown to have significant variability and large impact on the turbine response (Dimitrov et al., 2015; Downey, 2006; Eggers et al., 2003; Ernst et al., 2012 Holtstag et
al., 2016; Kelly et al., 2014; Matthaus et al., 2017; Moriarty et al., 2002; Rinker, 2016; Saraynsoontorn et al., 2008; Sathe et al., 2012; Sutherland, 2002; Wagner et al., 2010; Walter et al., 2009). The shear exponent, or wind profile, is the next most common parameter examined, concluded to have similar or slightly less importance to the turbulence intensity (Bulaevskaya et al., 2015; Dimitrov et al., 2015; Downey, 2006; Eggers et al., 2006; Ernst et all., 2012; Kelly et al., 2014; Matthaus et al., 2017; Sathe et al., 2012). Other parameters investigated include the turbulence length scale, standard deviation of different
directional wind components, Richardson number, spatial coherence, component correlation, and veer. Mixed conclusions are drawn on the importance of these secondary parameters, which are influenced by the range of variability considered (based on the conditions examined), the turbine control system, as well as the turbine size and hub height under consideration. The effects of considering the secondary wind parameters are also mixed, sometimes increasing and sometimes decreasing the loads in the turbine; however, most agree that the use of site-specific measurements of the wind parameters will lead to a more accurate
assessment of the turbine loads, resulting in designs that are either better optimized or lower risk.

The focus of this case study is to obtain a thorough assessment of which wind characteristics influence wind turbine structural loads when considering the variability of these parameters over a wide sampling of site conditions.

### 4.1.1 Parameters

A total of 18 input parameters were chosen to represent the wind-inflow conditions, considering the mean wind profile, velocity spectrum, spatial coherence, and component correlation, as summarized in Table 2. The parameters used were identified considering a Veers model for describing and generating the wind characteristics because it provides a quantitative description





with a known and limited set of inputs. Each of these parameters is described in the following sub-sections. Note that the Veers model differs from the other commonly used Mann turbulence model.[1] Regardless, the Veers model is used here because it is more tailorable than the Mann model, *i.e.*, there are more input parameters that can be varied.

**Table 2. Wind-inflow condition parameters (18 total).**

| Mean Wind Profile | Velocity Spectrum | Spatial Coherence | Component Correlation |
|---|---|---|---|
| Shear ($\sigma$) | Standard deviation, u ($\sigma_u$) | Input coherence decrement, u ($a_u$) | Reynolds stress, uw ($PC_{uw}$) |
| Veer ($\beta$) | Standard deviation, v ($\sigma_v$) | Input coherence decrement, v ($a_v$) | Reynolds stress, uv ($PC_{uv}$) |
| | Standard deviation, w ($\sigma_w$) | Input coherence decrement, w ($a_w$) | Reynolds stress, vw ($PC_{vw}$) |
| | Integral scale parameter, u ($L_u$) | Offset parameter, u ($b_u$) | |
| | Integral scale parameter, v ($L_v$) | Offset parameter, v ($b_v$) | |
| | Integral scale parameter, w ($L_w$) | Offset parameter, w ($b_w$) | |
| | | Exponent ($\gamma$) | |

**4.1.1.1 Mean Wind Profile**

A standard power-law shear model is used to describe the vertical wind speed profile and a linear wind direction veer model is used. The sensitivity of these characteristics are captured through variation of the exponent of the shear, $\alpha$, and the total veer
(in degrees) across the turbine, $\beta$ (centered around the hub, following right-hand-rule about the vertical axis of the turbine). The IEC 61400-1 standard (IEC, 2005) uses $\alpha = 0.2$ and $\beta = 0$ under normal turbulence.

**4.1.1.2 Velocity Spectrum**

The Veers model uses a Kaimal spectrum to represent the turbulence. The Kaimal spectrum is defined as (IEC, 2005):

$$\frac{f S_q(f)}{\sigma_q^2} = \frac{4 f L_q / V_{hub}}{(1 + 6 f L_q / V_{hub})^{5/3}} \tag{6}$$

where $f$ is the frequency (Hz), $q$ is the index of the velocity component direction ($u, v, w$), $S_q$ is the single-sided velocity spectrum, $V_{hub}$ is the mean wind speed at hub height, $\sigma_q$ is the velocity component standard deviation, and $L_q$ is the velocity component integral scale parameter. The IEC 61400-1 standard (IEC, 2005) uses a wind-speed-dependent standard deviation, *i.e.*, $\sigma_1 = 0.14(0.75 V_{hub} + 5.6 \text{ m/s})$, and a set scaling between the direction components of the standard deviation and scale
parameters, *i.e.*, $\sigma_2 = 0.8\sigma_1$; $\sigma_3 = 0.8\sigma_1$; $L_u = 8.1 \times (42 \text{ m}) = 340.2 \text{ m}$; $L_v = 2.7 \times (42 \text{ m}) = 113.4 \text{m}$; and $L_w = 0.66 \times (42 \text{ m}) = 27.72$ m. However, in this study each parameter in ($u, v, w$) is varied independently. An inverse Fourier transform is applied to the Kaimal spectrum and random phases derived from the turbulence seed to derive a turbulent time series for each of the wind components independently.

---

[1] The Mann turbulence model (also considered in the IEC 61400-1 standard) is based on a three-dimensional tensor representation of the turbulence derived from rapid distoration of isotropic turbulence using a uniform mean velocity shear (Jonkman, 2009). The Mann model considers the three turbulence components as dependent, representing the correlation between the longitudinal and vertical components resulting from the Reynolds stresses. In the IEC 61400-1 standard, the two spectra (Mann and Kaimal) are equated, resulting in three parameters that may be set for the Mann model. However, there is uncertainty in whether the loads resulting from these two different turbulence spectra are truly consistent.





### 4.1.1.3 Spatial Coherence Model

The point-to-point spatial coherence (*Coh*) quantifies the frequency-dependent cross-correlation of a single turbulence component at different transverse points in the wind inflow grid. The general coherence model used in TurbSim is defined as:

$$Coh_{q,i,j,f} = exp\left( -a_q \left(\frac{d}{z_m}\right)^\gamma \sqrt{\left(\frac{fd}{V_{hub}}\right)^2 + \left(b_q d\right)^2} \right) \tag{7}$$

where $d$ is the distance between points $i$ and $j$, $z_m$ is the mean height of the two points (IEC, 2005), and $V_{hub}$ is the mean wind speed at hub height. The variables $a_q$ and $b_q$ are the input coherence decrement and offset parameter, respectively. Note that the use of $V_{hub}$ in the general coherence model is a modification to the standard TurbSim method. The model is based on the IEC coherence model with the added term $(d/z_m)^\gamma$ – introduced by Solari (1987) – where $\gamma$ can vary between 0 and 1. The

IEC 61400-1 standard (IEC, 2005) does not use the $(d/z_m)^\gamma$ term and uses $a_u = 12$ and $b_u = 0.12/L_u$. Coherence is not defined in (IEC, 2005) for the transverse wind components $v$ and $w$.

### 4.1.1.4 Component Correlation Model

The component-to-component correlation (*PC*) quantifies the cross-correlation between directional turbulence components at a single point in space. For example, $PC_{uw}$ quantifies the correlation between the $u$ and $w$ turbulence components at a given

point. TurbSim modifies the $v$- and $w$-component wind speeds by computing a linear combination of the time series of the three independent wind speed components to obtain the mean Reynolds stresses ($PC_{uw}$, $PC_{uv}$, and $PC_{vw}$) at the hub. Note that because this calculation occurs in the time domain, the velocity spectra of the $v$- and $w$-components are somewhat affected by the enforced component correlations. The IEC 61400-1 standard (IEC, 2005) does not specify Reynolds stresses.

### 4.1.2 Parameter Ranges

To assess the sensitivity of each of the parameters on the load response, a range over which the parameters could vary was defined. The variation level was assessed through a literature search seeking the range over which the parameters could realistically vary for different wind-farm sites around the world (Berg et al., 2013; Bulaevskaya et al., 2015; Clifton; Dimitrov et al., 2016; Dimitrov et al., 2015; Eggers et al., 2003; Ernst et al., 2012; Holtslag et al., 2016; Jonkman, 2009; Kalverla et al., 2017; Kelley, 2011; Lindelöw-Marsden, 2009; Matthaus et al., 2017; Moriarty et al., 2002; Moroz, 2017; Nelson et al., 2003;

Park et al., 2015; Rinker, 2016; Saint-Geours et al., 2010; Saranyasontoorn et al., 2004; Saranyasontoorn et al., 2008; Sathe et al., 2012; Solari, 1987; Sutherland, 2002; Teunissen, 1970; Wagner et al., 2010; Walter et al., 2009; Wharton et al., 2015; Ziegler et al., 2016). When possible, parameter ranges were set based on wind speed bins. If no information on wind-speed dependence was found, the same values were used in all bins. The ranges, summarized in Table 3, were taken from multiple sources (references cited below the values), based on measurements across a variety of different locations and conditions. For

comparison, the nominal value prescribed by IEC for category $B$ turbulence is specific in the "Nom" row.

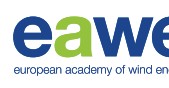
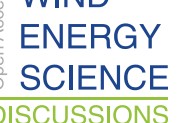


**Table 3: Included wind-inflow parameter ranges separated by wind speed bin.**

| | $\alpha$ | $\beta$ (deg) | $L_u$ (m) | $L_v$ (m) | $L_w$ (m) | $\sigma_u$ (m/s) | $\sigma_v$ (m/s) | $\sigma_w$ (m/s) | $a_u$ | $a_v$ | $a_w$ | $b_u$ | $b_v$ | $b_w$ | $\gamma$ | $PC_{uw}$ (m²/s²) | $PC_{uv}$ (m²/s²) | $PC_{vw}$ (m²/s²) |
|---|---|---|---|---|---|---|---|---|---|---|---|---|---|---|---|---|---|---|
| **Below-rated wind speed, 3-10 m/s** | | | | | | | | | | | | | | | | | | |
| Nom. | 0.2 | 0 | 340 | 110 | 28 | 1.6 | 1.3 | 1.3 | 12 | - | - | 3.5E-4 | - | - | 0 | - | - | - |
| Min. | -1.5* | -25 | 5 | 2 | 2 | 0.05 | 0.02 | 0.03 | 1.5 | 1.7 | 2 | 0 | 0 | 0 | 0 | -3.5 | -4.5 | -2.7 |
| Max. | 3.3 | 50 | 1,000 | 1,000 | 650 | 7.2 | 7.4 | 4.5 | 26 | 18 | 17 | 0.08 | 4.5E-3 | 0.011 | 1 | 0.50 | 6.0 | 1.0 |
| Ref. | Clifton | Walter et al., 2009 | Nelson et al., 2003; Solari et al., 2001 | Nelson et al., 2003 | Nelson et al., 2003; Teunissen, 1970 | Clifton | Clifton | Clifton | Solari 1987 | Saranyasoontorn et al., 2006 | Solari 1987 Solari et al., 2001 | Saranyasoontorn et al., 2004 | Jonkman 2009 | Jonkman 2009 | Solari 1987 | Kelley 2011 | Kelley 2011 | Kelley 2011 |
| **Near-rated wind speed, 10-14 m/s** | | | | | | | | | | | | | | | | | | |
| Nom. | 0.2 | 0 | 340 | 110 | 28 | 2.0 | 1.6 | 1.6 | 12 | - | - | 3.5E-4 | - | - | 0 | - | - | - |
| Min. | -0.4 | -10 | 8 | 2 | 2 | 0.20 | 0.05 | 0.05 | 1.5 | 1.7 | 2 | 0 | 0 | 0 | 0 | -3.5 | -4.5 | -2.7 |
| Max. | 0.9 | 50 | 1,400 | 1,300 | 450 | 7.3 | 8.1 | 4.3 | 26 | 18 | 17 | 0.08 | 3.0E-3 | 6.0E-3 | 1 | 0.50 | 6.0 | 1.0 |
| Ref. | Dimitrov et al., 2015; Moroz 2017 | Dimitrov et al., 2015; Walter et al., 2009 | Nelson et al., 2003; Solari et al., 2001 | Nelson et al., 2003; | Nelson et al., 2003; | Clifton; Kelley 2011 | Clifton; Kelley 2011 | Clifton; Kelley 2011; Nelson et al., 2003 | Solari 1987 | Saranyasoontorn et al., 2006 | Saranyasoontorn et al., 2006; Solari 1987 | Saranyasoontorn et al., 2004 | Jonkman 2009 | Jonkman 2009 | Solari 1987 | Kelley 2011 | Kelley 2011 | Kelley 2011 |
| **Above-rated wind speed, 14-25 m/s** | | | | | | | | | | | | | | | | | | |
| Nom. | 0.2 | 0 | 340 | 110 | 28 | 2.7 | 2.1 | 2.1 | 12 | - | - | 3.5E-4 | - | - | 0 | - | - | - |
| Min. | -0.4 | -10 | 25 | 2 | 2 | 0.20 | 0.18 | 0.15 | 1.5 | 1.7 | 2 | 0 | 0 | 0 | 0 | -3.5 | -4.5 | -2.7 |
| Max. | 0.7 | 25 | 1,600 | 1,500 | 650 | 7.4 | 7.3 | 4.2 | 26 | 18 | 18 | 0.05 | 2.5E-3 | 6.5E-3 | 1 | 0.50 | 6.0 | 1.0 |
| Ref. | Dimitrov et al., 2015; Moroz 2017 | Dimitrov et al., 2015; Walter et al., 2009 | Kelly et al., 2014; Nelson et al., 2003 | Nelson et al., 2003 | Nelson et al., 2003 | Clifton Kelley 2011; Nelson et al., 2003; | Clifton Kelley 2011; Nelson et al., 2003 | Clifton Kelley 2011; | Solari 1987 | Saranyasoontorn et al., 2006 | Jonkman 2009; Solari et al., 2001 | Saranyasoontorn et al., 2004 | Jonkman 2009 | Jonkman 2009 | Solari 1987 | Kelley 2011 | Kelley 2011 | Kelley 2011 |

\* This value was changed to -0.75 due to simulation issues

To simplify the screening of the most influential parameters, all parameters were considered independent of one another. This was done because of the difficulty of considering correlations between a large number of parameters. Such correlations should be studied in future work once parameter importance has been established. Since each parameter was considered independently, except for the conditioning on wind speed bin, some unphysical parameter combinations may arise. This was considered acceptable for the screening process.

### 4.1.3 Elementary Effects

The EE value was calculated for each of the 18 parameters ($I$) at 30 different points ($R$) in the input-parameter hyperspace. The number of points was determined through a convergence study on the average of the EE value. At each of the points examined, $S$ different turbulent wind files (i.e. $S$ independent time-domain realizations from $S$ seeds) were run. Thirty seeds were needed based on a convergence study of the ultimate and fatigue load metrics for all QoIs. Based on these numbers, the total number of simulations performed for the wind-inflow case study was $R \times (I+1) \times S \times B = 30 \times 19 \times 30 \times 3 = 51,300$, where $B$ is the number of wind speed bins considered.

The EE values across all input parameters, input hyperspace points, and wind speed bins were examined for each of the QoIs for ultimate and fatigue loads. To identify the most sensitive parameters, a tally was made of the number of times an EE value



exceeded the threshold for a given QoI. The resulting tallies are shown in Figure 3, with the ultimate load tally on the left and the fatigue load tally on the right. As expected, these plots show an overwhelming level of sensitivity of the u-direction turbulence standard deviation ($\sigma_u$) and also the vertical wind shear ($\alpha$). However, focusing on the lower tally values in this plot (shown in Figure 4) highlights the secondary-level of importance of veer ($\beta$), u-direction integral length ($L_u$) and components of the IEC coherence model ($a_u$ and $b_u$), as well as the exponent ($\gamma$).

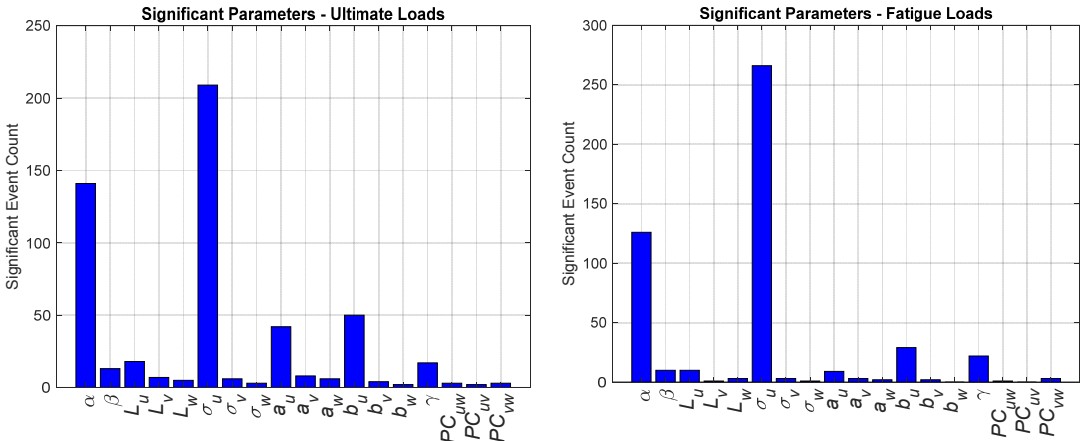

**Figure 3: Identification of significant parameters using ultimate (left) and fatigue (right) loads. Significant events are defined by number of outliers identified across each of the QoIs for all wind speed bins, input parameters, and simulation points.**

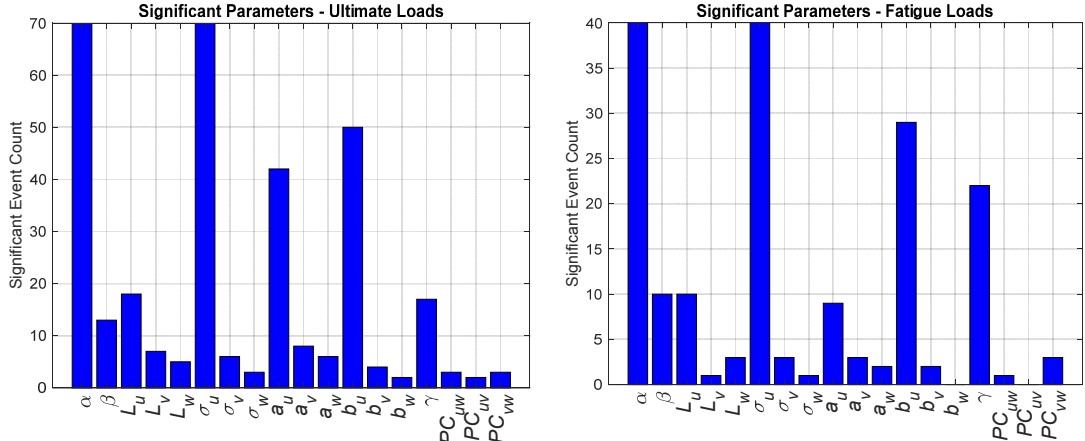

**Figure 4: Zoomed in view of identification of significant parameters using ultimate (left) and fatigue (right) loads.**

Exceedance probability and histograms of the EE values for each of the QoIs are plotted in Figure 5 – Figure 8 for the ultimate and fatigue load metrics. Each plot contains all calculated EE values for a given QoI colored by wind speed bin. The threshold used to identify significant EE values is shown in each plot as a solid black line. All points above the threshold line indicate a significant event and are included in the outlier tally for each QoI. Note that while electric power is shown, it is not used in the



outlier tally because its variation is strictly limited by the turbine controller rather than other wind parameters. Highlighted in these figures is that most of the outliers come from the below-rated wind speed bin.

To understand why the below-rated wind speed bin would be creating the most outliers, a more thorough examination is made for one of the QoIs. Exceedance probability plots of blade-root loads are shown in Figure 9. Here, all input parameters are plotted independently of each other to compare the behavior between parameters. Each line represents a different input parameter, with each point representing a different location in the hyperspace. These plots show how the shear and u-component standard for the lower wind speed bin stand-out compared to all other parameters; likewise, the u-component standard deviation stands out across different wind speed bins for the ultimate load. One of the reasons that the shear value

shows such a large sensitivity in the lowest wind speed bin is the large range over which the parameter is varied. A smaller range is used for the near- and above-rated bins, resulting in less sensitivity to shear for those wind speeds. The impact of the range on the sensitivity of the parameter indicates that for sites with extreme conditions, such as an extreme shear, using appropriate parameter values in a loads analysis can be important in accurately assessing the ultimate and fatigue loading on the turbine. The effect of shear could also be diminished by employing independent blade-pitch control, whereas the reference

NREL 5-MW turbine controller used here employs collective blade-pitch control.

Histogram plots of blade-root bending moment EE values are shown in Figure 10 and Figure 11. In each figure, wind speed bins are displayed in different plots and EE value histograms showing the contribution from all input parameters are shown in each histogram. Ultimate load EE values are shown in Figure 10 and fatigue load EE values are shown in Figure 11.

Highlighted in these plots is the large sensitivity of the shear parameter and, to a lesser extent, u-component standard deviation in the far extremes.




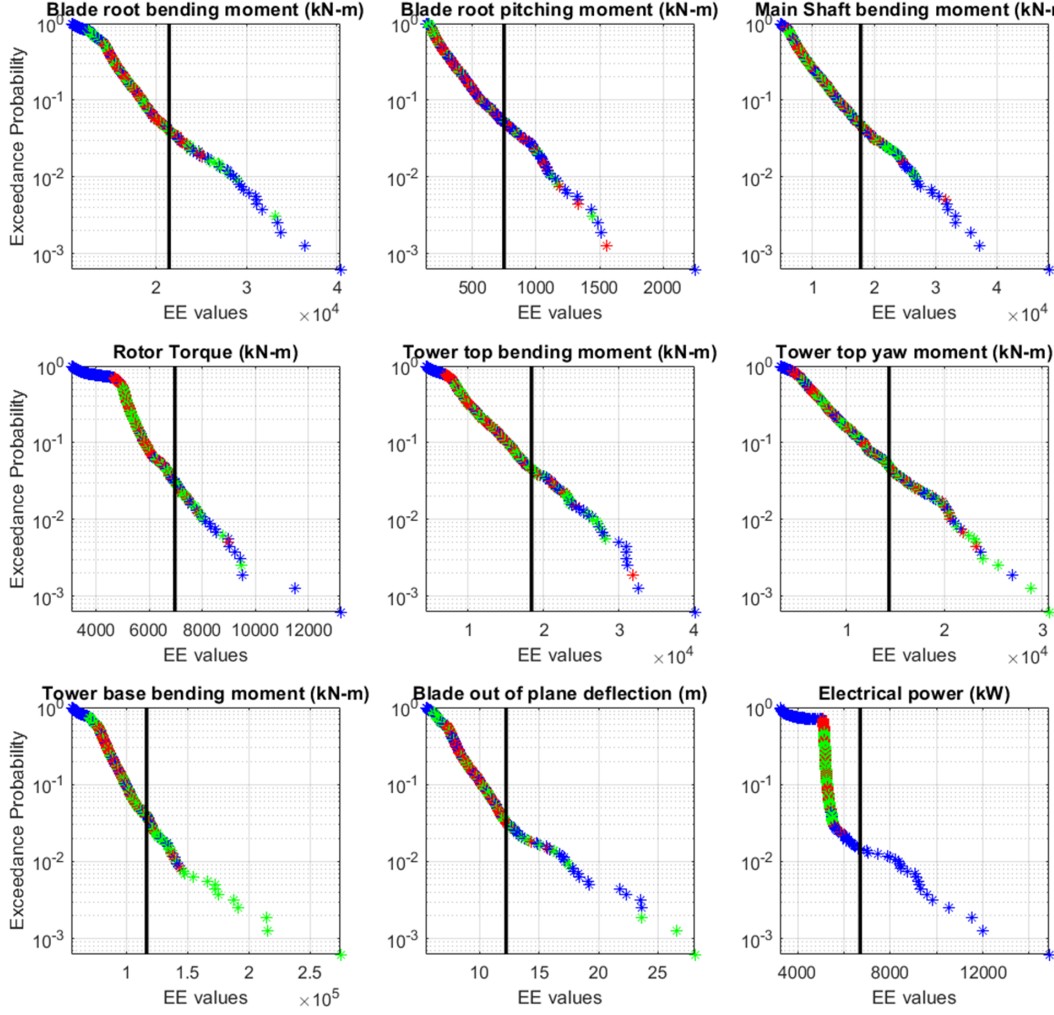

**Figure 5: Exceedance probability plot of ultimate load EE values for each of the QoIs across all wind speed bins, input parameters, and simulation points. Black line represents the defined threshold by which outliers are counted for each QoI. Color indicates wind speed bin (blue=below rated, red=near rated, green=above rated).**

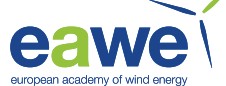
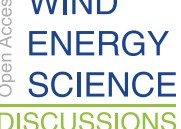

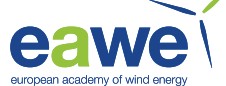

**Figure 6: Exceedance probability plot of fatigue load EE values for each of the across all wind speed bins, input parameters, and simulation points. Black line represents the defined threshold by which outliers are counted for each QoI. Color indicates wind speed bin (blue=below rated, red=near rated, green=above rated).**





**Figure 7: Stacked histogram of the ultimate load EE values for each of the QoIs across all wind speed bins, input parameters, and simulation points. Black line represents the defined threshold by which outliers are counted for each QoI. Color indicates wind speed bin (blue=below rated, red=near rated, green=above rated).**




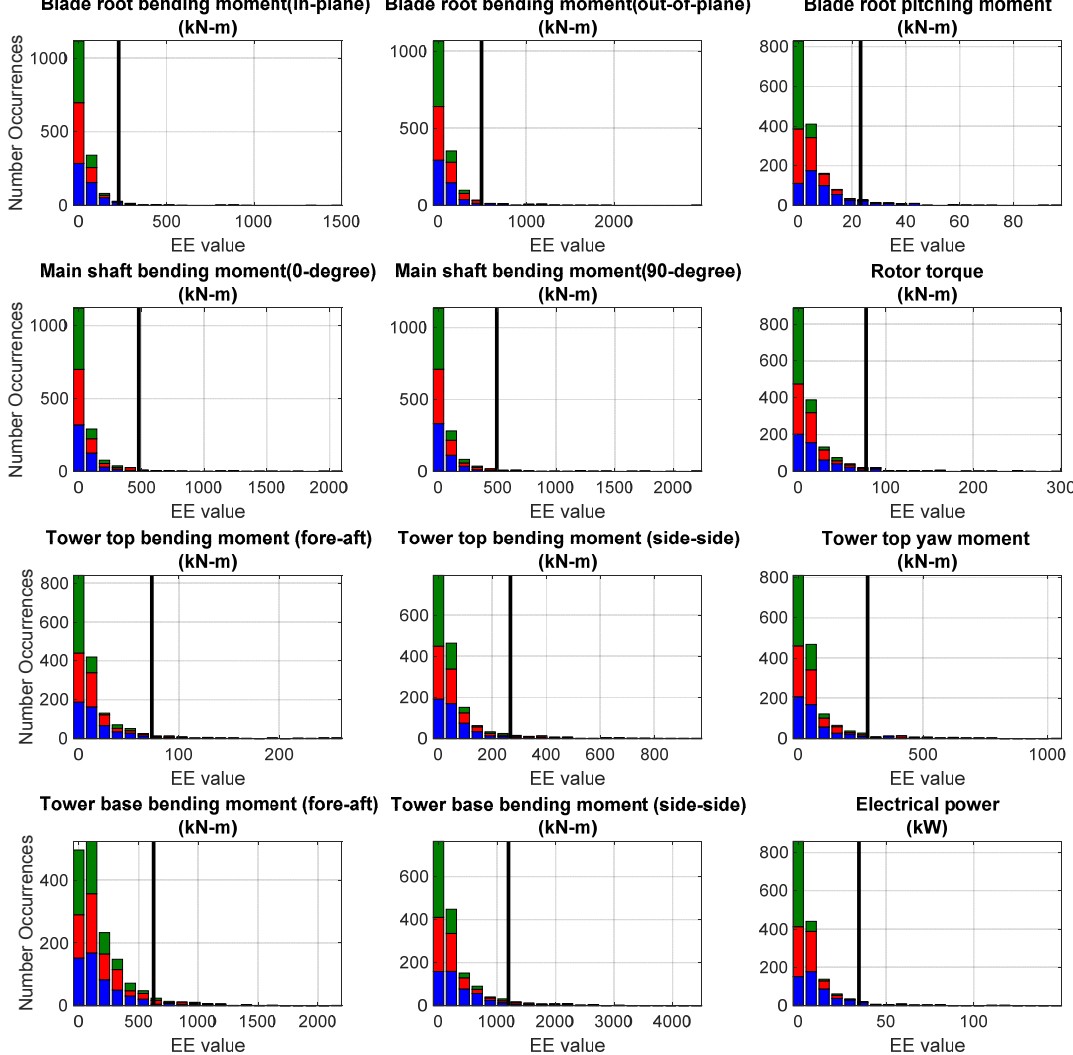

Figure 8: Stacked histogram of the fatigue load EE values for each of the QoIs across all wind speed bins, input parameters, and simulation points. Black line represents the defined threshold by which outliers are counted for each QoI. Color indicates wind speed bin (blue=below rated, red=near rated, green=above rated).

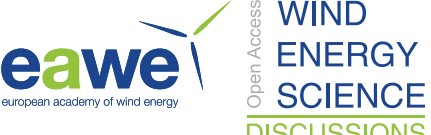


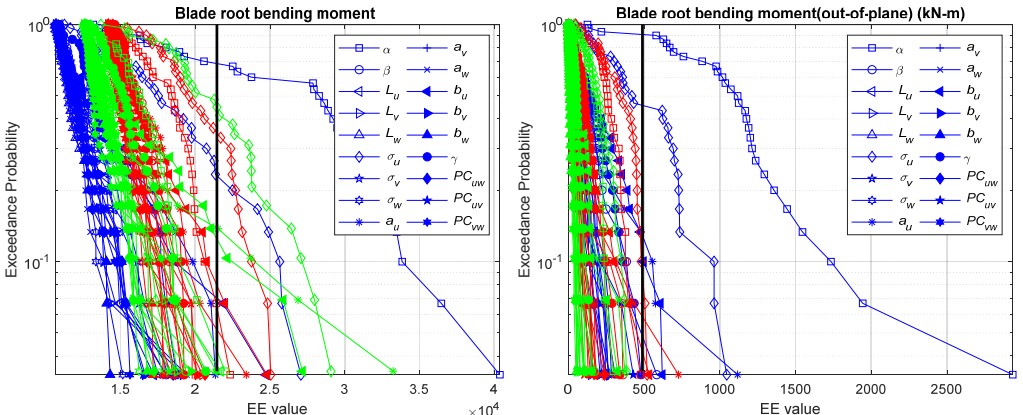

**Figure 9: Exceedance probability plot of ultimate (left) and fatigue (right) load EE values for blade-root bending moments. Each line represents a different input parameter and wind speed bin (blue=below rated, red=near rated, green=above rated).**

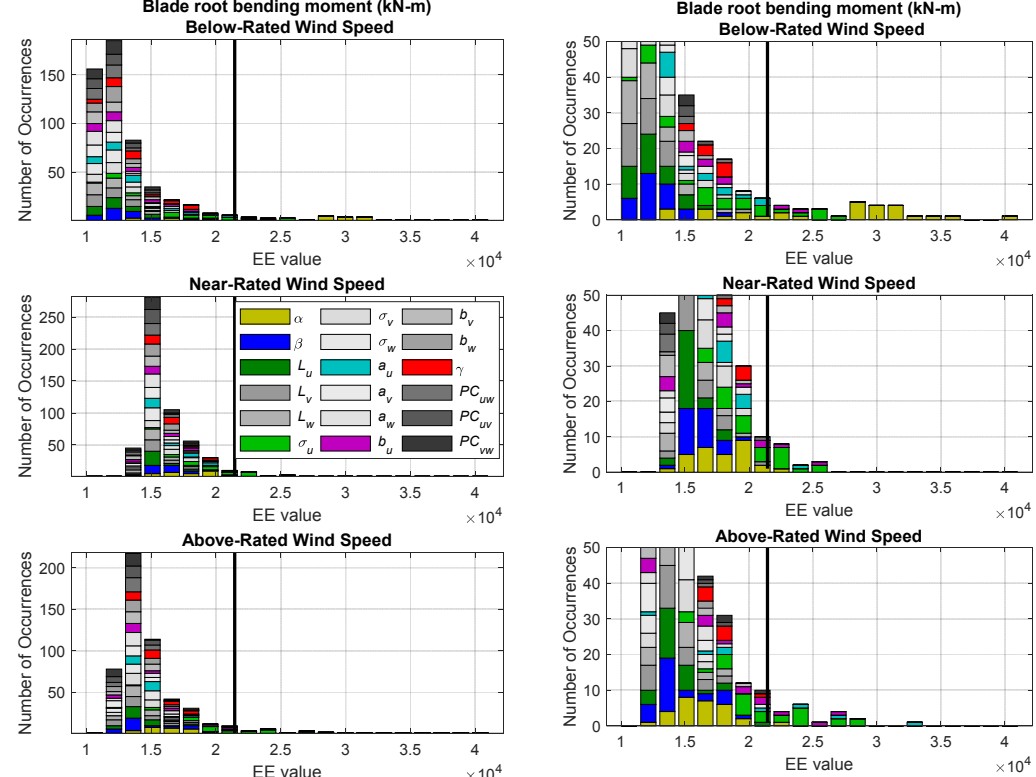

**Figure 10: Histogram of ultimate load EE values for the blade-root bending moment. Each graph (in left column) shows one wind speed bin and includes all input parameters. Right column is a zoom of the left.**




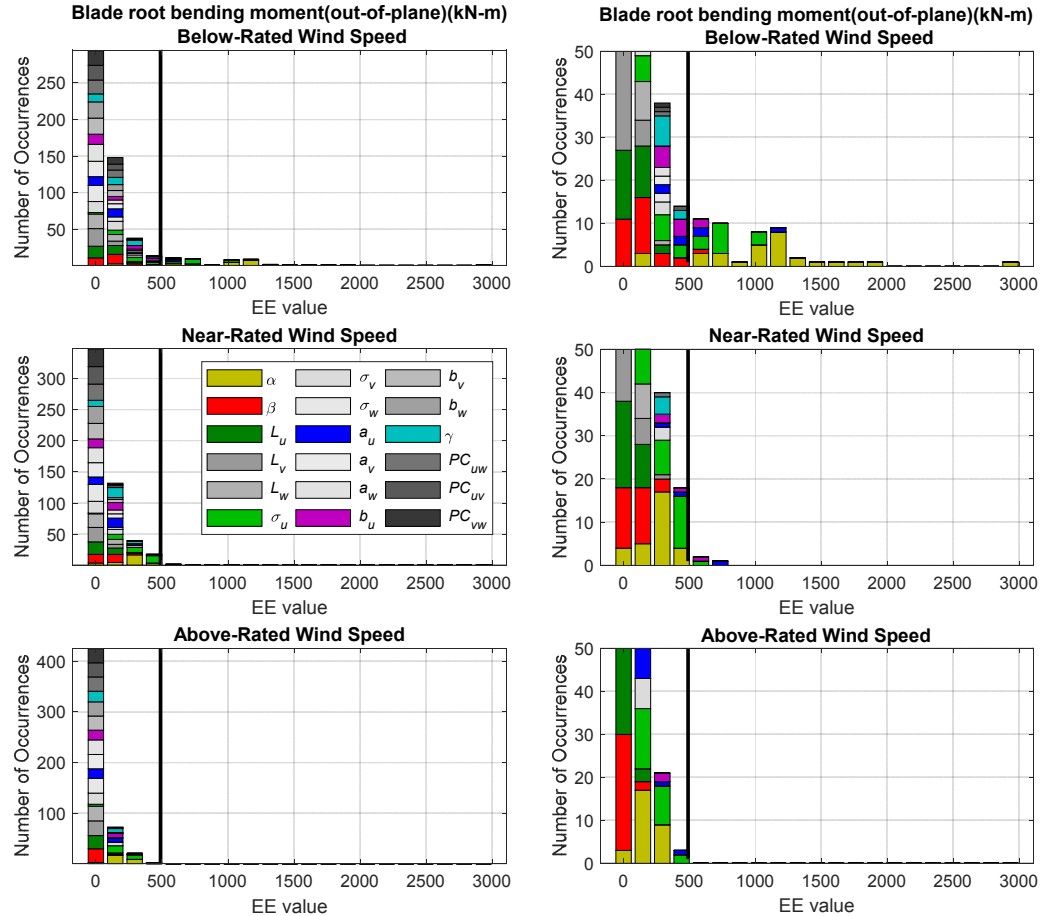

**Figure 11: Histogram of fatigue load EE values for the blade-root bending out-of-plane bending moment. Each graph (in left column) shows one wind speed bin and includes all input parameters. Right column is a zoom of the left.**

To summarize which parameters are important for which QoIs, the number of times each input parameter contributed to the significant event count for a given QoI was tallied. The top most sensitive parameters are shown in Table 4 and Table 5 for
5  ultimate and fatigue loads, respectively. Overall, 46% of the outliers for both ultimate and fatigue loads are due to u-direction turbulence standard deviation ($\sigma_u$), and 26% for vertical shear ($\alpha$), and for all but two outputs, these are the most sensitive parameters. The two exceptions are blade-root pitching moment and tower-base bending moment, which show u-direction turbulence standard deviation as the most important parameter, but show coherence properties and integral scale parameter as more important than shear. This is understandable since shear will have little effect on collective blade pitching and rotor
10  thrust. The remaining parameters have far less significance, with only components of the IEC coherence model, $a_u$ (5%) and $b_u$ (8%), having a value great than 1%. These results can be used in future sensitivity analysis work to focus on perturbation of specific input parameters based on desired turbine loads.





**Table 4. Top input parameters contributing to ultimate load sensitivity of each QoI.**

| Blade-Root Bending Mom. | Blade-Root Pitch Mom. | Main Shaft Bending Mom. | Rotor Torque | Tower-Top Bending Mom. | Tower-Top Yaw Mom. | Tower-Base Bending Mom. | Blade OoP Deflection |
|---|---|---|---|---|---|---|---|
| $\sigma_u$ (29) | $\sigma_u$ (26) | $\alpha$ (33) | $\sigma_u$ (28) | $\alpha$ (31) | $\sigma_u$ (41) | $\sigma_u$ (22) | $\alpha$ (22) |
| $\alpha$ (22) | $\gamma$ (10) | $\sigma_u$ (28) | $\alpha$ (12) | $\sigma_u$ (22) | $\alpha$ (10) | $a_u, b_u, L_u$ (8) | $\sigma_u$ (13) |
| $b_u$ (7) | $L_u$ (8) | $a_u, b_u$ (5) | $b_u$ (6) | $a_u, b_u$ (5) | $a_u, b_u$ (8) | -- | $b_u$ (9) |

**Table 5. Top input parameters contributing to fatigue load sensitivity of each QoI.**

| Blade-Root IP Bend. Moment | Blade-Root OP Pitch. Moment | Blade-Root Pitch. Moment | Main Shaft Bending Moment 0° | Main Shaft Bending Mom. 90° | Rotor Torque | Tower-Top FA Bend. Moment | Tower-Top SS Bend. Moment | Tower-Top Yaw Moment | Tower-Base FA Bend. Moment | Tower-Base SS Bend. Moment |
|---|---|---|---|---|---|---|---|---|---|---|
| $\sigma_u$ (10) | $\alpha$ (14) | $\sigma_u$ (14) | $\alpha$ (18) | $\alpha$ (18) | $\sigma_u$ (25) | $\sigma_u$ (31) | $\sigma_u$ (47) | $\sigma_u$ (48) | $\sigma_u$ (24) | $\sigma_u$ (35) |
| $\alpha$ (8) | $\sigma_u$ (9) | $L_u$ (7) | $\sigma_u$ (12) | $\sigma_u$ (11) | $\alpha$ (11) | $\alpha$ (12) | $\alpha$ (13) | $\alpha$ (12) | $\alpha$ (6) | $\alpha$ (9) |
| $b_u$ (3) | $b_u$ (4) | $\alpha$ (5) | $\beta$ (2) | $\beta$ (2) | $b_u$ (9) | $b_u$ (7) | $\gamma$ (3) | $\gamma$ (3) | $\gamma$ (4) | $\gamma$ (4) |

5 **4.2 Aeroelastic Turbine Properties**

The second case study focuses on which aeroelastic turbine parameters have the greatest influence on turbine ultimate and fatigue loads during normal turbine operation. These properties are categorized into four main categories: support structure, blade structural, blade aerodynamic, and controller properties.

10 It is widely acknowledged that uncertainty in the aerodynamic parameters can affect the prediction of turbine performance and structural loading (Abdallah et al., 2015; Madsen et al., 2010; Simms et al, 2001). Abdallah et al. (2015) demonstrated the impact of uncertainty in steady airfoil data on prediction of extreme loads and assessed the correlation between various static coefficient polars (Abdallah et al., 2015). Despite significant work to measure these parameters, considerable uncertainty remains in their prediction. Static lift and drag measurements almost exclusively come from wind tunnel tests of airfoils, which 15 lack three-dimensional and unsteady effects that are instead estimated through the application of semi-empirical engineering models, e.g., rotational augmentation (stall delay) and stall hysteresis (Abdallah et al., 2015, Simms et al, 2001). In Damiani (2010), unsteady aerodynamic parameters were tuned for several airfoil sections to match experimental lift and drag unsteady hysteresis loops, but the consequences of parameter variation were not considered. Blade chord and twist ranges were chosen using the work of Loeven and Bihl (2008), who identified changes in blade chord and twist based on uncertainty in 20 aerodynamic loading, icing, or wear of the blades.

Beyond the blade aerodynamic properties, other turbine properties also contribute to the uncertainty of the load response characteristics. Abdallah (2018) provides a comprehensive assessment of the sources of uncertainty affecting the prediction of loads in a wind turbine. Researchers have not focused on these other parameters as significantly as the aerodynamic ones, but 25 they could have a significant contribution to the uncertainty. Witcher (2017) examined uncertainty in properties such as the tower and blade mass/stiffness properties within the context of defining a probabilistic approach to designing wind turbines by examining distributions of the load from propagated input parameter uncertainties versus resistance distributions. Prediction of the reliability of the wind turbine has been studied through examination of the damping in the structure by Koukoura (2014) and a better understanding of the uncertainty in the properties of the drivetrain by Holierhoek (2010). Limited information is 30 available on what the actual ranges of uncertainty are for these different characteristics. For most studies, expert opinion is used to set a realistic bound. A better assessment of these bounds will be needed in future work to understand the relative importance of the physical parameters and to provide a more precise assessment of the uncertainty bounds in the load response of wind turbines.





### 4.2.1 Parameters

For the turbine aeroelastic properties, 39 input parameters were identified covering support structure properties, blade structural properties, blade aerodynamic properties (both steady and unsteady characteristics), and controller properties. These parameters are summarized in Table 6 (acronyms are defined in the following subsections).

### 4.2.2 Parameter Ranges

The level of variation was based on the perceived level of uncertainty in the parameter values. Some of these levels of uncertainty are proposed within the literature, but when no other information was available, expert opinion was used. The source for the information is provided below the values in each table summarizing the parameter ranges. "Exp" is used to identify where expert opinion was used. The uncertainty levels are largely percentage based, but in some instances an exact
value was used. The following sub-sections define the ranges of the parameters introduced in Table 6. All parameters were considered independent of one another, as was done for the wind parameter sensitivity analysis.

**Table 6. Turbine aeroelastic parameters (39 total).**

| Support Structure Properties | Blade Structural Properties | Blade Aerodynamic Properties | Controller Properties |
|---|---|---|---|
| Nacelle mass ($N_{Mass}$) | Blade flapwise stiffness ($B_{FK}$) | Twist (∅) at tip | Yaw Angle Error ($\theta$) |
| Nacelle CM x-location ($N_{CM}$) | Blade edgewise stiffness ($B_{EK}$) | Chord ($c$) at root and tip | Collective pitch error ($∅_{err,coll}$) |
| Tower CM location ($T_{CM}$) | Blade flapwise stiffness imbalance ($B_{FK,imb}$) | Leading-edge separation time constant ($T_{f0}$) | Imbalanced pitch error ($∅_{err,imb}$) |
| Tower stiffness ($T_{KF}$) | Blade edgewise stiffness imbalance ($B_{EK,imb}$) | Vortex shedding time constant ($T_{V0}$) | |
| Tower mass ($T_{MD}$) | Blade damping ratio ($B_{DR}$) | Leading-edge pressure gradient time constant ($T_p$) | |
| Tower damping ratio ($T_{DR}$) | Blade mass ($B_M$) | Vortex advection time constant ($T_{VL}$) | |
| Drivetrain stiffness ($D_K$) | Blade mass imbalance ($B_{M,imb}$) | Strouhal Number ($St_{sh}$) | |
| Drivetrain damping ($D_D$) | Blade CM location ($B_{CM}$) | Lift ($C_l$) at root and tip | |
| Shaft angle ($\alpha_S$) | Precone ($\beta_p$) | TES Lift AoA ($\alpha_{TES}$) at root and tip | |
| | | Max Lift AoA ($\alpha_{max}$) at root and tip | |
| | | SR Lift AoA ($\alpha_{SR}$) at root and tip | |
| | | 0-degree drag ($C_{d,0}$) at root and tip | |

### 4.2.2.1 Support Structure Properties

**Table 7: Parameter value ranges of turbine support structure parameter ranges.**

| | $N_{mass}$ (kg) | $N_{CM}$ (m) | $T_{CM}$ (m) | $T_{KF}$ (-) | $T_{MD}$ (-) | $T_{DR}$ (%) | $D_K$ $\left(\frac{N-m}{rad}\right)$ | $D_D$ $\left(\frac{N-m}{rad/sec}\right)$ | $\alpha_S$ (deg) |
|---|---|---|---|---|---|---|---|---|---|





| | | | | | | | | | |
|---|---|---|---|---|---|---|---|---|---|
| **Nom.** | 240,000 | 1.9 | 42.505 | 1.02 | 1 | 2.55 | 867,637,000 | 6,215,000 | -5 |
| **Min.** | 216,000 | 1.71 | 40.38 | 0.72 | 0.95 | 0.1 | 780,873,300 | 0.0 | -5.2 |
| **Max.** | 264,000 | 2.09 | 44.63 | 1.32 | 1.05 | 5.0 | 954,400,700 | 12,430,000 | -4.8 |
| **Ref.** | Witcher, 2017 | Exp | Exp | Koukoura 2014 | Witcher, 2017 | Koukoura 2014 | Holierhoek et al., 2010 | Holierhoek et al., 2010 | Santos and van Dam, 2015 |

#### 4.2.2.2 Blade Structural Properties

For the blade structural properties, 9 parameters were considered, including blade flapwise and edgewise stiffness (including stiffness imbalance), mass (including mass imbalance), CM, damping, and precone angle, as detailed in Table 8. Through ElastoDyn, blade structural dynamics are modeled using two flapwise mode shapes and one edgewise mode shape per blade.

To manipulate blade structural response, the frequency of the flapwise and edgewise mode shapes was changed by ±5% of 0.7 Hz and 1 Hz, respectively, by uniformly scaling the associated stiffness. The blade mass was changed by uniformly scaling the distributed blade mass of all blades. The nominal scaling of 1.04536 is described in the NREL 5-MW specifications document (Jonkman et al., 2009). The blade CM location was changed by varying the blade root and tip density such that density increased at one end and decreased at the other without changing the overall blade mass. Blade imbalance effects were

also included by varying the mode frequency and mass of each blade. The imbalances were introduced by applying a different change value to each blade. Specifically, one blade is modified to be a value that is higher than the nominal value, and another modified to a lower value. The third blade remains unchanged at the nominal value.

**Table 8: Parameter value ranges of turbine blade structure parameter ranges.**

| | $B_{FK}$ (-) | $B_{EK}$ (-) | $B_{FK,imb}$ (-) | $B_{EK,imb}$ (-) | $B_{DR}$ (%$_{critical}$) | $B_M$ (-) | $B_{M,imb}$ (-) | $B_{CM}$ (m) | $\beta_p$ (deg) |
|---|---|---|---|---|---|---|---|---|---|
| **Nom.** | 1 | 1 | 0.01 | 0.01 | 1.55 | 1.04536 | 0.025 | 0.015 | -2.5 |
| **Min.** | 0.9 | 0.9 | 0.0 | 0.0 | 0.1 | 0.993 | 0.0 | 20.60 | -2.75 |
| **Max.** | 1.1 | 1.1 | 0.02 | 0.02 | 3.0 | 1.1 | 0.05 | 22.60 | -2.25 |
| **Ref.** | Exp | Exp | Exp | Exp | Exp | Witcher, 2017 | Exp | IEC, 2010 | Exp |

#### 4.2.2.3 Blade Aerodynamic Properties

The blade aerodynamic properties were represented using 18 parameters: 3 associated with blade twist and chord distribution; 10 associated with the static aerodynamic component; and 5 associated with the unsteady aerodynamic properties.

Blade twist and chord distributions were manipulated by specifying a change in the distributions along the blade. Three parameters were defined, associated with changing the chord at the blade tip and root, and the twist at the blade tip. For each

of these parameters, the associated distribution along the blade was modified linearly such that there was zero change at the opposite end. The root twist was not changed because the blade-pitch angle uncertainties are considered in the controller parameter section.

#### 4.2.2.4 Steady Airfoil Aerodynamics

For the steady aerodynamic component, the lift and drag versus angle-of-attack (AoA) curves were modified to examine the

sensitivity on resulting loads throughout the wind turbine. The turbine operated in normal operating conditions, and therefore only relevant regions of the curves were modified. To modify the curves, the curves were parameterized using an approach based on one introduced by Abdallah (2015). The approach used here parameterizes the $C_l$ and $C_d$ curves using five points; these points were perturbed and a spline fit to the points. The points of interest are:

- Beginning of linear $C_l$ region — determines the lower limit of the AoA range of interest and was kept constant ($\alpha_{lin}$,

$C_{l,lin}$);




- $C_d$ value at AoA = 0° (**0°, $C_{d,0}$**);
- Trailing edge separation (TES) point — AoA location at which $C_l$ curve is no longer linear ($\boldsymbol{\alpha_{TES}}$, $\boldsymbol{C_{l,TES}}$);
- AoA location at which $C_l$ reaches a maximum ($\boldsymbol{\alpha_{max}}$, $\boldsymbol{C_{l,max}}$);
- Separation reattachment (SR) point — AoA location at which slope of $C_l$ curve is no longer negative ($\boldsymbol{\alpha_{SR}}$, $\boldsymbol{C_{l,SR}}$).

The selected points of interest are similar to those selected by Abdallah (2015). A notable difference is the consideration of $C_{d,0}$ as opposed to $C_{d,90}$, which is the $C_d$ value at a $\alpha$=90°. $C_{d,0}$ was chosen for this study because of the focus on normal operational region, as opposed to the extreme conditions considered by Abdallah (2015). The three variable points of interest were perturbed by a percentage of the default value. The perturbations and correlations are depicted in Figure 12 and parameter ranges are detailed in Table 9. From Abdallah (2018), the TES, max, and SR Cl values for an individual airfoil have a

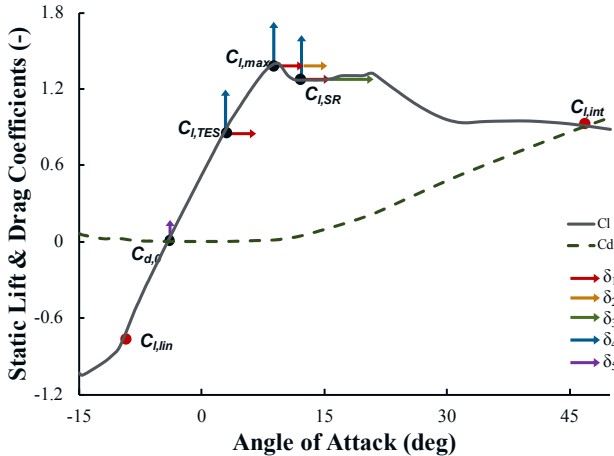

**Figure 12: Perturbation of points of interest in representative $C_l$ and $C_d$ curves.**

correlation to one another of 0.9. Thus, all Cl values are perturbed collectively, using the same percentage (δ4). The AoA values are less correlated and are therefore perturbed independently of one another. However, to ensure that nonphysical relative values are not reached, all AoA values are perturbed by the same base percentage (δ1), and then an additional independent variation of a smaller value was added (δ2 and δ3) for $\alpha_{max}$ and $\alpha_{SR}$, respectively. The $C_{d,0}$ value was also perturbed (δ5).

$C_l$ and $C_d$ curves were altered for each airfoil. However, instead of specifying $\delta$ values for each airfoil, these values were specified at the root and tip airfoils, excluding the cylindrical airfoils at the base. Perturbation values for the interior airfoils were computed from a linear fit of the endpoint values. The method of developing the new curves for each airfoil is detailed here:

    1.   AoA deltas are applied to the original AoA values via the following equations:

$$\boldsymbol{\alpha_{TES,new} = \alpha_{TES,orig} + \alpha_{TES,orig}\delta_1} \tag{8}$$
$$\boldsymbol{\alpha_{max,new} = \alpha_{max,orig} + \alpha_{max,orig}(\delta_1 + \delta_2)} \tag{9}$$
$$\boldsymbol{\alpha_{SR,new} = \alpha_{SR,orig} + \alpha_{SR,orig}(\delta_1 + \delta_3)} \tag{10}$$



2. The new AoA values are fit to the nearest existing AoA value on the curve. The AoA-value resolution is fine enough that all perturbations are captured, though not precisely. This approach may need to be adjusted if the perturbations were to decrease.

3. For all new AoA values, the change in $C_l$ between the original $C_l$ value ($C_{l,orig}$) and the $C_l$ curve value at the new AoA ($C_{l,orig+}$) is computed via:

$$\epsilon = C_{l,orig+} - C_{l,orig}$$

4. The total change in $C_l$ is then computed via:

$$C_{l,diff} = \delta_4 C_{l,orig} - \epsilon$$

This ensures that if $\delta_4 = 0$, the final $C_{l,new}$ value is equivalent to that of the original curve.

5. For $C_l$ perturbation, the end points are located at the AoA associated with the beginning of the linear $C_l$ region ($C_{l_{lin}}$) and AoA = 90°; as these are fixed points, they have $C_{l,diff} = 0$. The $C_l$ curve is replaced by a line between ($\alpha_{lin}$, $C_{l_{lin}}$) and ($\alpha_{TES}$, $\delta_4 C_{l_{TES}}$). A piece-wise linear spline – representing perturbations about the original curve – is constructed between the points ($\alpha_{TES_{new}}$, $\delta_4 C_{l_{TES}}$); ($\alpha_{max_{new}}$, $\delta_4 C_{l_{max}}$); ($\alpha_{SR_{new}}$, $\delta_4 C_{l_{SR}}$); and (90°, 0°).

6. The $C_{l,diff}$ values calculated from the spline fit are added to the original $C_l$ curve.

$$C_{l,new} = C_{l,orig+} + C_{l,diff}$$

A similar process was followed by modifying the $C_d$ curves, wherein the $C_d$ value corresponding to $\alpha = 0°$ ($C_{d,0}$) is perturbed by a specified value ($\delta_5$) in the same manner as the $C_l$ values. A piece-wise linear spline is then fit between (-90°, $C_{d,-90}$), (0°, $C_{d,0}$), and (90°, $C_{d,90}$) and added to the original $C_d$ curve. $C_{d,0}$ is constrained to not go below 0.

Several modified $C_l$ and $C_d$ curves for each airfoil section are shown in . Note that $C_d$ curves are perturbed, but by a very small amount not visible in the plots. These perturbations result in modified $C_l$ and $C_d$ curves that maintain the primary characteristics of the original curve, but differ in both magnitude and feature location.

### 4.2.2.5 Unsteady Airfoil Aerodynamics

There are several unsteady airfoil aerodynamic parameters that can be modified in OpenFAST. By expert opinion (Damiani, 2018), several of these parameters have been identified as having the largest potential variability or impact on turbine response and are therefore included in this study. Several of the parameters in the Beddoes-Leishman-type unsteady airfoil aerodynamics model used here are derivable from the (perturbed) static lift and drag polars (i.e., when the lift and drag polars are perturbed, the associated Beddoes-Leishman unsteady airfoil aerodynamic parameters are perturbed as well). Additionally, there are several other parameters associated with unsteady aerodynamics that are included in OpenFAST. These parameters are:

- $T_{f0}$ — time constant connected to leading-edge separation of the airfoil;
- $T_{V0}$ — time constant connected to vortex shedding;
- $T_p$ — time constant connected to the leading-edge pressure gradient;
- $T_{VL}$ — time constant connected to the vortex advection process;
- $St_{sh}$ — Strouhal number associated with the vortex shedding frequency.

These quantities were varied over the ranges detailed in Table 9 and are constant across the blade.

**Table 9: Parameter value ranges of turbine blade aerodynamic parameter ranges.**

| | $\varnothing_{tip}$ (deg) | $c_r$ (m) | $c_t$ (m) | $T_{f0}$ (-) | $T_{V0}$ (-) | $T_p$ (-) | $T_{VL}$ (-) | $St_{sh}$ (-) | $C_{l,tr}$ (-) | $\alpha_{TES,tr}$ (deg) | $\alpha_{max,tr}$ (deg) | $\alpha_{SR,tr}$ (deg) | $C_{d,0,tr}$ (-) |
|---|---|---|---|---|---|---|---|---|---|---|---|---|---|
| **Nom.** | 0.106 | 3.542 | 1.419 | 6.5 | 8 | 1.35 | 16.5 | 0.245 | Var. | Var. | Var. | Var. | Var. |
| **Min.** | -1.894 | 3.1878 | 1.2771 | 3 | 1 | 1 | 11 | 0.19 | -26% | -20% | -8% | -15% | -100% |
| **Max.** | 2.106 | 3.8962 | 1.5609 | 10 | 15 | 1.7 | 22 | 0.3 | +26% | +20% | +8% | +15% | +100% |





| Ref. | Petrone et al., 2011 | Loeven and Bijl, 2008 | Loeven and Bijl, 2008 | Damiani et al., 2016 | Damiani et al., 2016 | Damiani et al., 2016 | Damiani et al., 2016 | Damiani et al., 2016 | Abdallah et al., 2015 | Abdallah et al., 2015 | Abdallah et al., 2015 | Abdallah et al., 2015 | Ehrmann et al., 2017 |
|------|------|------|------|------|------|------|------|------|------|------|------|------|------|

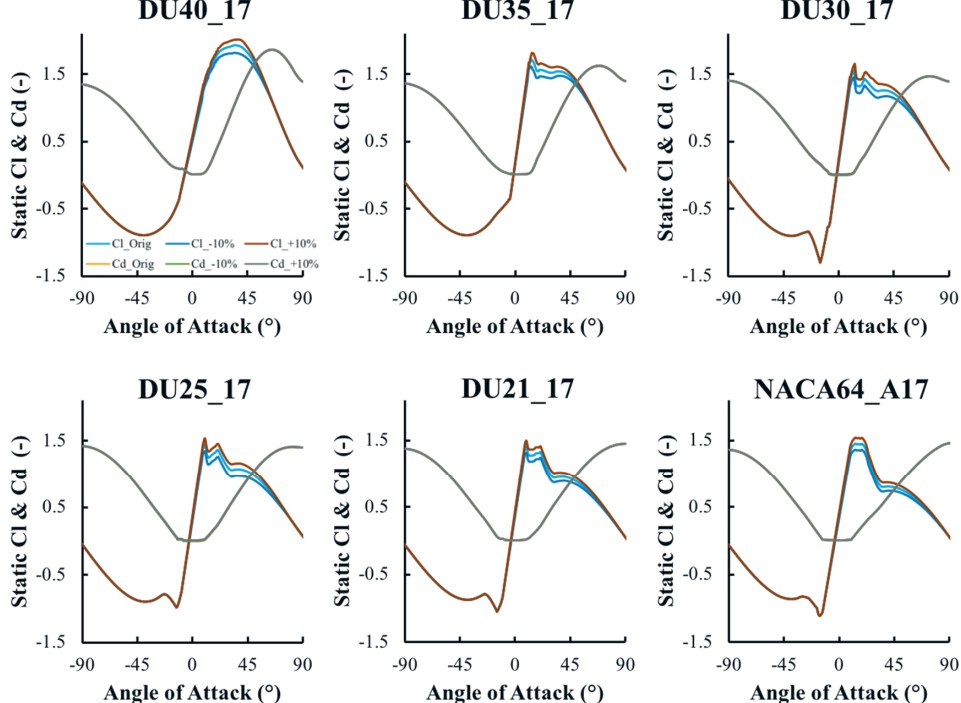

**Figure 13: Sample original and perturbed $C_l$ and $C_d$ curves for each airfoil section used in the NREL 5MW reference turbine. Perturbed values represent ±10% of the specified range for each parameter.**

5   **4.2.2.6 Controller Properties**

Turbine yaw error was incorporated by directly changing the yaw angle of the turbine. For the collective blade pitch error, the twist distribution of each blade is identically shifted uniformly along the blade independent of the twist change in Table 9. For the imbalance pitch error, modified twist distributions are applied to two of the blades: one with a higher-than-nominal tip twist, one with a lower-than-nominal tip twist, and one unchanged.

**Table 10: Parameter value ranges of turbine controller parameter ranges.**

|        | $\theta$ (deg) | $\emptyset_{err,coll}$ (deg) | $\emptyset_{err,imb}$ (deg) |
|--------|------|------|------|
| **Nom.** | 0 | 0 | 0.1 |
| **Min.** | -20 | -0.2 | 0 |
| **Max.** | 20 | 0.2 | 0.2 |



| **Ref.** | Quick et al., 2017 | Simms et al., 2001 | Simms et al., 2001 |
|---|---|---|---|

### 4.3 Elementary Effects

The EE value calculation and analysis process are the same as was used for the wind parameter analysis. 60 wind file seeds were needed based on a convergence study of the ultimate and fatigue load metrics for all QoIs. This increase in the number in required wind file seeds over the other study is likely due to some turbine input parameter combinations causing resonance.
Based on these numbers, the total number of simulations performed for the wind-inflow case study was $R \times (I+1) \times S \times B = 30 \times 40 \times 60 \times 3 = 216{,}000$.

The EE values across all input parameters, input hyperspace points, and wind speed bins were examined for all QoI for the ultimate and fatigue loads. For each QoI, the number of times an EE value exceeded the threshold for a given QoI was tallied.
The resulting tallies are shown in Figure 14, with the ultimate load tally on the top and fatigue load tally on the bottom. Note that nearly twice as many significant events were counted for fatigue loads; less significant events were counted for ultimate loads because of the limited threshold exceedance in the below-rated wind speeds. The percentage that each relevant input parameter contributed to the total significant event count is summarized in Table 11. Ultimate turbine loads are most sensitive to yaw error ($\theta$) and lift ($C_l$) distribution at the outboard section of the blade ($C_{l,t}$), which combined accounted for nearly half
of all significant events. Fatigue loads are also highly sensitive to blade mass imbalance ($B_{M,imb}$). Turbine loads are also sensitive to twist distribution ($\phi$), blade mass ($B_M$), and the $C_l$ distribution at the inboard section of the blade ($C_{l,b}$). Though these results are expected, their relevant importance is likely a new finding. Other input parameters that were found to affect turbine load sensitivity are inboard maximum AoA ($\alpha_{max,b}$), blade mass center of mass ($B_{CM}$), blade flapwise stiffness ($B_{FK}$), nacelle center of mass location ($N_{CM}$), nacelle mass ($N_{mass}$), chord length at the inboard section of the blade ($c_b$), tower stiffness
($T_{KF}$), drivetrain damping ($D_D$), and inboard trailing edge separation AoA ($\alpha_{TES,b}$). The AoA values at the inboard section of the blade are likely more important than at the outboard section due to the higher likelihood of the inboard section operating with higher AoA near stall. Additionally, the range of AoA values at the inboard section is larger than at the outboard section because the nominal inboard AoA values are higher, which could contribute to greater sensitivity.





**Figure 14: Identification of significant parameters using ultimate (top) and fatigue loads (bottom). Significant events are defined by the number of outliers identified across each of the QoIs for all wind speed bins, input parameters, and simulation points.**





**Table 11. Percentage of contribution to total number of significant events for ultimate and fatigue loads.**

|  | $\theta$ (deg) | $B_{M,imb}$ (-) | $C_{l,t}$ (-) | $\emptyset$ (deg) | $B_M$ (-) | $C_{l,b}$ (-) | $\alpha_{max,b}$ (deg) | $B_{CM}$ (-) | $B_{FK}$ (-) | $N_{CM}$ (m) | $N_{mass}$ (kg) | $\alpha_{TES,b}$ (deg) | $c_b$ (m) | $T_{KF}$ (m) | $D_D$ (m) |
|---|---|---|---|---|---|---|---|---|---|---|---|---|---|---|---|
| **Ult. Load (%)** | 21.5 | 8.8 | 21.9 | 12.7 | 2.5 | 9.2 | 4.0 | 0.6 | 3.6 | 3.0 | 2.9 | 0.8 | 1.9 | 0.2 | 1.3 |
| **Fat. Load (%)** | 23.7 | 21.2 | 17.4 | 8.8 | 10.4 | 6.7 | 1.9 | 3.7 | 0.3 | 0.0 | 0.6 | 1.9 | 0.6 | 1.7 | 1.4 |

Exceedance probability of the EE values for all QoIs are plotted in Figure 15 and Figure 16 for the ultimate and fatigue load metrics, respectively. The figures were prepared in the same manner as Figure 5 and Figure 6. As shown in Figure 15, ultimate
load EE values are largely grouped by wind speed bin for all QoIs with the exception of blade-root pitching moment. These results highlight the unequal distribution of outliers resulting from each turbine QoI. Most notably, blade-root pitching moment accounts for 18% of the total ultimate load significant events, whereas rotor torque accounts for only 5%. This suggests that it may be better to tailor the threshold for each QoI, but this was deemed overly complicated for this first pass at assessing the sensitivity. Additionally, for a given QoI, it is typical for all ultimate load significant events to occur from either the near- or
above-rated wind speeds. However, fatigue load EE values are more evenly distributed across wind speed bins, as shown in Figure 16. The lower significant event counts for ultimate loads is a result of the segregated nature of the ultimate load EE values, as opposed to the more evenly distributed nature of the fatigue load EE values. In fact, unlike ultimate load EE values, a large percentage of significant events result from below-rated wind speed cases because of the higher probability of low wind speed conditions. However, the distribution of fatigue load outliers resulting from each turbine QoI is approximately the
same as the distribution for ultimate load outliers, with 14.6% of outliers resulting from the blade-root OoP bending moment and only 4.3% resulting from the blade-root pitching moment. Note that the QoI (blade-root pitching moment) that contributed the most outliers for ultimate load outliers contributes the least for fatigue load outliers.

These points are further highlighted by computing EE value histograms for ultimate and fatigue load QoIs. Ultimate load
histograms are shown in Figure 17. Here, EE values are again colored by wind speed and the black vertical line represents the threshold for each QoI. The sharp separation of ultimate load EE values between wind speed bins is again evident. A zoomed-in view of the lower count values is shown in Figure 18. These plots highlight the separation of wind speed bin EE values. The more evenly distributed nature of the fatigue load EE values is further highlighted in the histogram plots depicted in Figure 19 and zoomed in Figure 20. Unlike ultimate load EE values, all wind speed bins contribute to the outlier count for each QoI.
Histogram plots of blade-root ultimate and fatigue bending moment EE values are shown in Figure 21 and Figure 22, respectively. The sharp separation of ultimate load EE values between wind speed bins is again evident. Highlighted in the fatigue load plots is the more even distribution of threshold-exceeding EE values across wind speed bins.





**Figure 15: EE-value exceedance probability plots of ultimate loads across all wind speed bins, input parameters, and simulations points for all QoIs. The black line represents the defined threshold by which outliers are counted for each QoI. Color indicates wind speed bin (blue=below rated, red=near rated, green=above rated).**





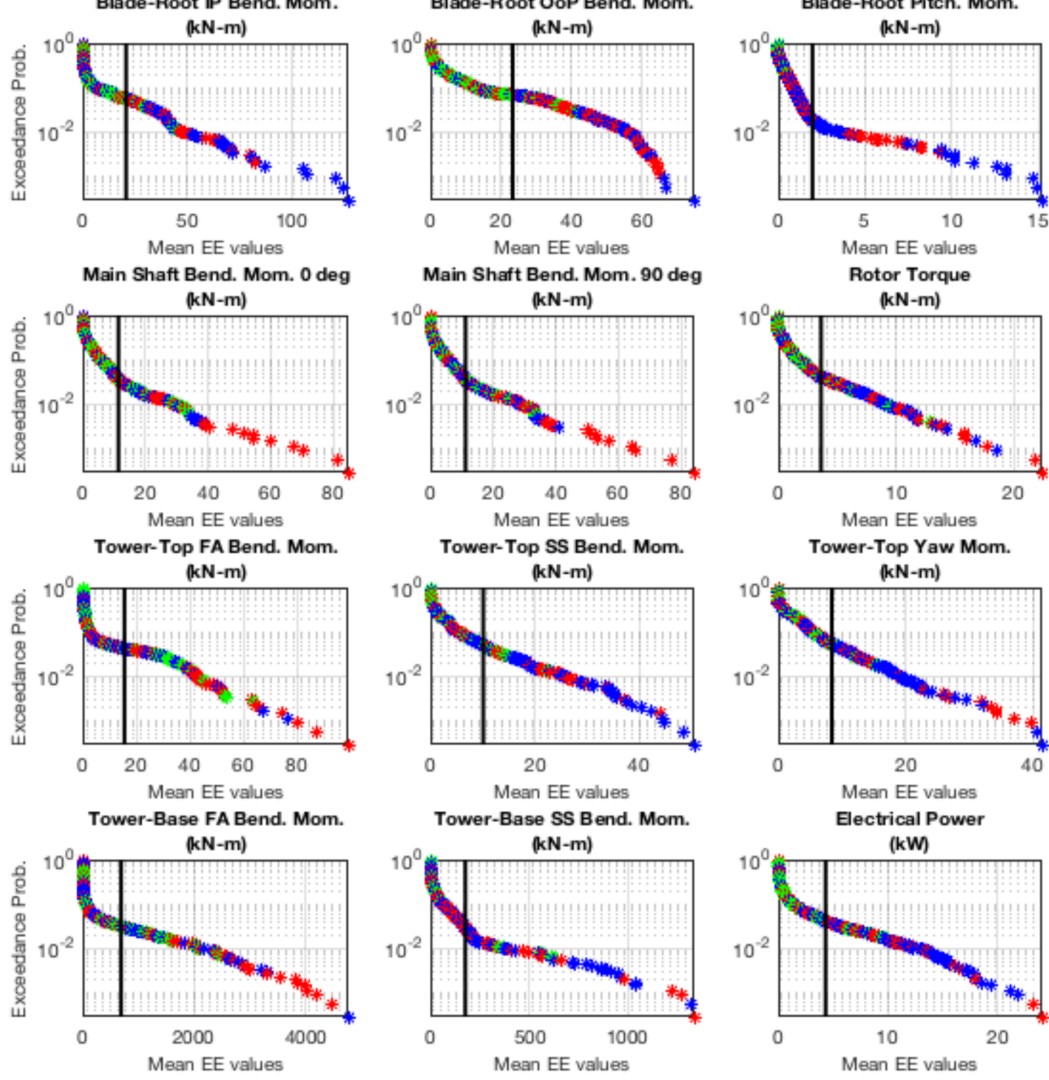

**Figure 16: EE-value exceedance probability plots of fatigue loads across all wind speed bins, input parameters, and simulations points for all QoIs. The black line represents the defined threshold by which outliers are counted for each QoI. Color indicates wind speed bin (blue=below rated, red=near rated, green=above rated).**



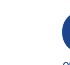

**Figure 17:** Stacked EE-values histograms of ultimate loads across all wind speed bins, input parameters, and simulation points for all QoIs. The black line represents the threshold by which outliers are counted for each QoI. Color indicates wind speed bin (blue=below rated, red=near rated, green=above rated).






**Figure 18: Zoomed-in stacked EE-values histograms of ultimate loads across all wind speed bins, input parameters, and simulation points for all QoIs. The black line represents the threshold by which outliers are counted for each QoI. Color indicates wind speed bin (blue=below rated, red=near rated, green=above rated).**




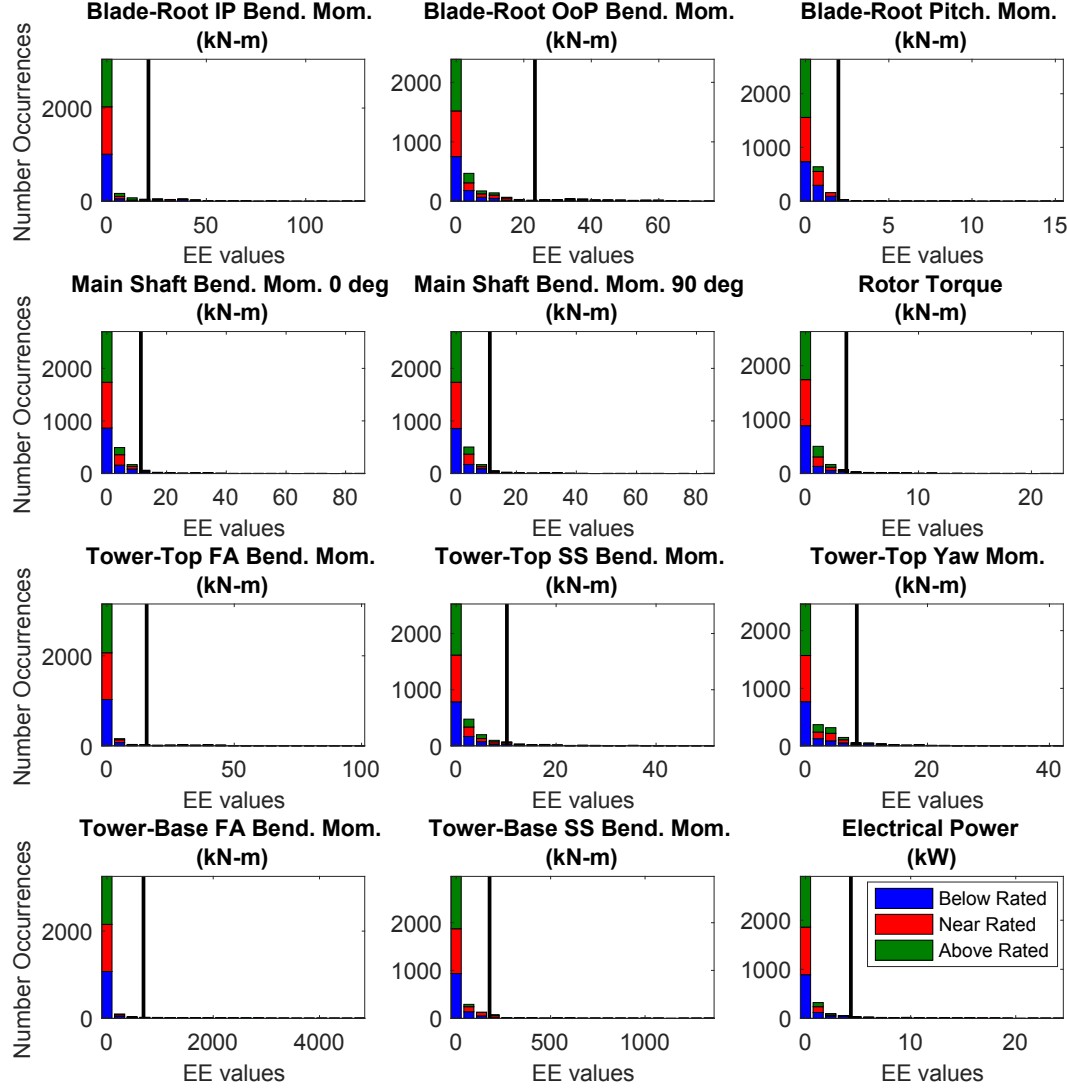

**Figure 19: Stacked EE-values histograms of fatigue loads across all wind speed bins, input parameters, and simulation points for all QoIs. The black line represents the threshold by which outliers are counted for each QoI. Color indicates wind speed bin (blue=below rated, red=near rated, green=above rated).**



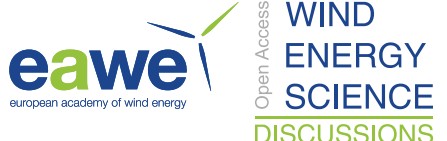

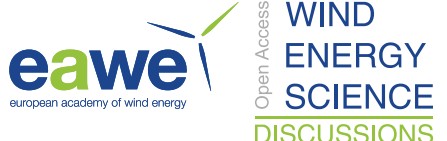

**Figure 20: Zoomed-in stacked EE-values histograms of fatigue loads across all wind speed bins, input parameters, and simulation points for all QoIs. The black line represents the threshold by which outliers are counted for each QoI. Color indicates wind speed bin (blue=below rated, red=near rated, green=above rated).**

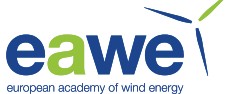
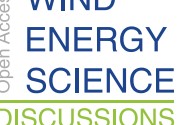

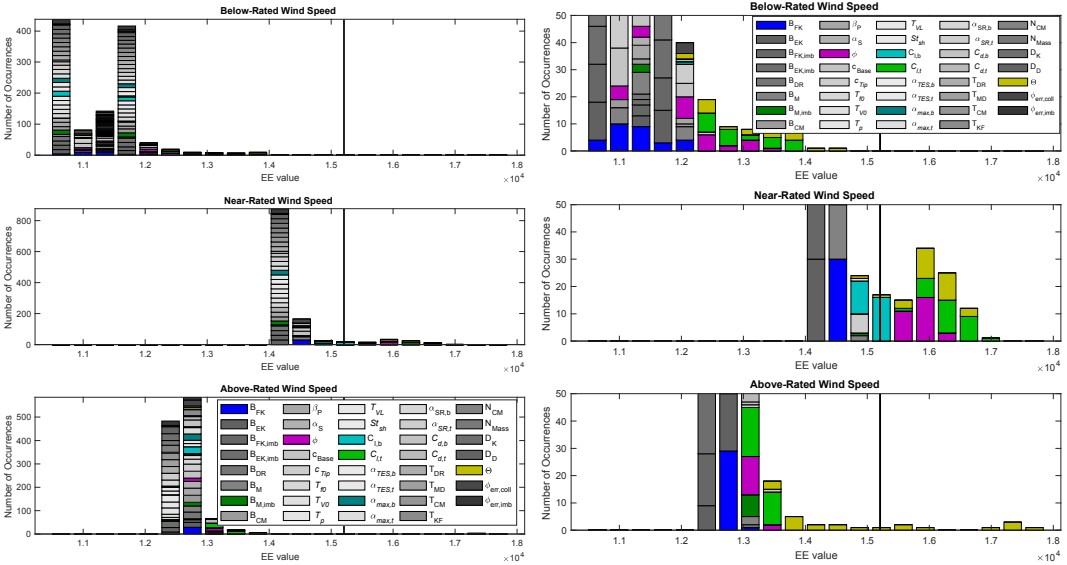

**Figure 21:** EE-values histograms of blade-root bending ultimate moment. Each graph shows one wind speed bin and includes all input parameters. Right column is a zoomed-in view of the left column.

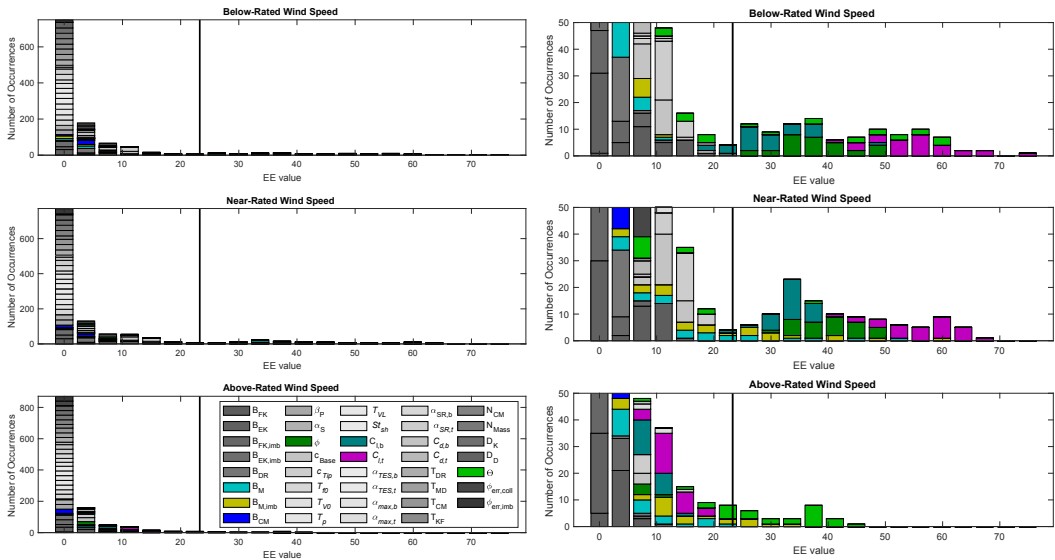

**Figure 22:** EE-value histograms of blade-root OoP bending fatigue moment. Each graph shows one wind speed bin and includes all input parameters. Right column is a zoomed-in view of the left column.

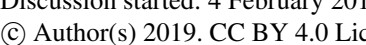
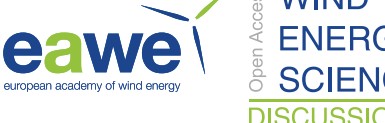


The behavior of blade-root loads are examined in more detail by plotting exceedance probability distinctly for each input parameter in Figure 23. Highlighted in these plots is the contribution of the individual input parameters to the outlier counts. For blade-root bending ultimate moment EE values, blade twist and $C_{l,t}$ EE values in the near-rated wind speed bin are beyond the threshold for every point in the hyperspace. Yaw error and $C_{l,b}$ EE values from the near-rated wind speed bin and yaw error
from the above-rated wind speed bin also cross the threshold. For blade-root OoP bending fatigue moment EE values, the threshold is exceeded by blade twist and $C_{l,t}$ EE values from the below- and above-rated wind speeds for every point in the hyperspace. However, for all other relevant input parameters, only certain points in the hyperspace result in threshold exceedance. This indicates that, for certain loads and input parameters, the sensitivity of the turbine is dependent on the combination of turbine parameter values. These results can be used in future studies to more thoroughly investigate the
hyperspace to determine how input parameter value combinations contribute to turbine sensitivity.

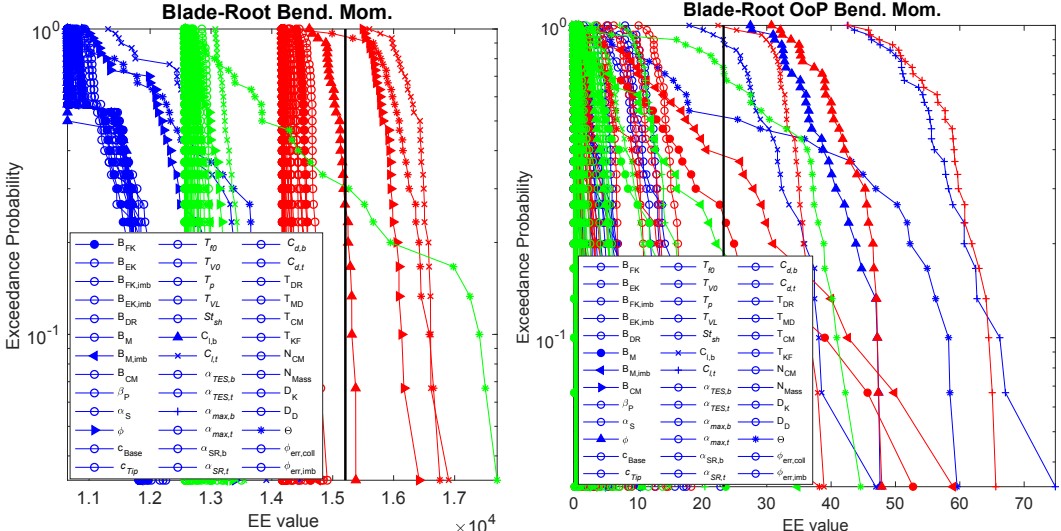

**Figure 23: EE-value exceedance probability plots for the blade-root bending ultimate moment (left) and blade-root OoP bending fatigue moment (right). Each line represents a different input parameter and wind speed bin (blue=below rated, red=near rated,**
**green=above rated).**

For each QoI, the number of times each input parameter contributed to the significant event count was tallied. The top parameters are shown in Table 10 and Table 11 for ultimate and fatigue loads, respectively. Overall, 63% of the top sensitive parameters for both ultimate and fatigue loads are due to aerodynamic perturbations or yaw error. Blade-root and main shaft moments are especially sensitive to perturbations of inputs. However, blade mass imbalance and blade mass account for 44%
of the most sensitive parameters associated with tower moment fatigue loads. Rotor torque ultimate and fatigue loads are most sensitive to perturbation of structural input parameters, especially those related to blade mass. For both ultimate and fatigue loads, electrical power is most sensitive to blade mass imbalance, blade mass factor, and yaw error. These results can be used in future sensitivity analysis work to focus on perturbation of specific input parameters based on desired turbine loads.

**Table 12. Top input parameters contributing to ultimate load sensitivity of each QoI.**

| Blade-Root Bend. Mom. | Blade-Root Pitch. Mom. | Main Shaft Bend. Mom. | Rotor Torque | Tower-Top Bend. Mom. | Tower-Top Yaw Mom. | Tower-Base Bend. Mom. | Blade OoP Deflection |
|---|---|---|---|---|---|---|---|
| $\theta$ (39) | $\alpha_{max,b}$ (33) | $\theta$ (43) | $B_{M,imb}$ (26) | $\theta$ (28) | $C_{l,t}$ (28) | $\phi/C_{l,t}$ (30) | $\phi/C_{l,b}$ (30) |


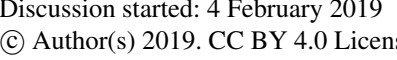


| $\phi/C_{l,t}$ (30) | $C_{l,b}$ (15) | $C_{l,t}$ (37) | $D_D$ (25) | $N_M$ (19) | $\theta$(25) | $C_{l,b}$ (28) | $B_{FK}$ (28) |
|---|---|---|---|---|---|---|---|
| $C_{l,b}$ (7) | $\phi$(14) | $c_b$ (22) | $B_M$ (17) | $N_{CM}/C_{l,t}$ (17) | $C_{l,b}$ (21) | $B_{M,imb}$ (27) | $\theta$(27) |

**Table 13. Top input parameters contributing to fatigue load sensitivity of each QoI.**

| Blade-Root IP Bend. Moment | Blade-Root OP Pitch. Moment | Blade-Root Pitch. Moment | Main Shaft Bending Moment 0° | Main Shaft Bending Mom. 90° | Rotor Torque | Tower-Top FA Bend. Moment | Tower-Top SS Bend. Moment | Tower-Top Yaw Moment | Tower-Base FA Bend. Moment | Tower-Base SS Bend. Moment |
|---|---|---|---|---|---|---|---|---|---|---|
| $B_{CM}$ (59) | $\phi/C_{l,t}$ (60) | $\alpha_{max,b}$ (31) | $\theta$(49) | $\theta$(49) | $B_{M,imb}$ (51) | $\theta$(75) | $\theta$(64) | $C_{l,t}$ (59) | $B_{M,imb}$ (59) | $T_{KF}$ (28) |
| $B_{M,imb}$ (52) | $C_{l,b}$ (54) | $\alpha_{TES,b}$ (18) | $C_{l,t}$ (44) | $C_{l,t}$ (42) | $\theta$(31) | $B_{M,imb}$ (54) | $C_{l,t}$ (52) | $B_{M,imb}$ (42) | $B_M$ (48) | $B_{M,imb}$ (26) |
| $B_M$ (47) | $\theta$(36) | $c_b$ (11) | $\phi$(16) | $\phi$(19) | $B_M$ (28) | $B_M$ (21) | $B_{M,imb}$ (38) | $\theta/\phi$(27) | $T_{DR}$ (2) | $B_M/C_{l,t}/N_M$ (10) |

**Conclusions**

A screening analysis of the most sensitive turbulent wind and aeroelastic parameters to the resulting structural loads and power
output was performed for the representative NREL 5-MW wind turbine under normal operating conditions. The purpose of
the study was to assess the sensitivity of different turbulent wind and turbine parameters on the resulting loads of the wind
turbine. The sensitivities of the different parameters were ranked. The study did not consider specific site conditions, but rather
focused on understanding the most sensitive parameters across the range of possible values for a variety of sites.

To limit the number of simulations required, a screening analysis using the EE method was used instead of a more
computationally intensive sensitivity analysis. The EE method is an assessment of the local sensitivity of a parameter at a
given location in space through variation of only that parameter, examined over multiple points throughout the parameter
hyperspace, making it a global sensitivity analysis. This work modified the general EE formula to examine the sensitivity of
parameters across multiple wind speed bins. A radial version of the method was employed, using Sobol numbers as starting
points, and a set delta value of 10% for the parameter variations.

Two independent case studies were performed. For the wind parameter case study, it was found that the loads and power are
highly sensitive to the shear and turbulence levels in the $u$-direction. To a lesser extent, turbine loads are sensitive to the wind
veer and the integral length scale and coherence parameters in the $u$-direction. The combinations of parameters in this study
spanned the ranges of several different locations. The parameters were considered independent of one another (conditioned
only on wind speed bin), which likely resulted in some non-physical wind scenarios. However, the screening analysis has
shown which parameters are most important to examine in more detail in future work.

The aeroelastic parameter case study showed that the loads and power are highly sensitive to the yaw error and the lift
distribution at the outboard section of the blade. To a lesser extent, turbine loads are sensitive to blade twist distribution, lift
distribution at the inboard section of the blade, and blade mass factor imbalance. Additionally, ultimate load EE values are
typically separated by wind speed bin, whereas fatigue load EE values are more evenly distributed across wind speed bins.

Through the implemented EE method, different combinations of input parameters have been used. When specific input
parameters are shown to be sensitive to one or more turbine loads, it is possible that only certain combinations of the input
parameters will result in this sensitivity. This leads to opportunities for future work to further investigate which parameter
combinations lead to higher turbine sensitivity. In future work, this ranking of most-sensitive parameters could be used to help
establish error bars around predictions of engineering models during validation efforts and provide insight into probabilistic
design methods and site-suitability analysis. While the most-sensitive ranking results may depend on the turbine size or



configuration, the analysis process developed hear could be applied universally to other turbines. This work could also be further expanded in future work to include load cases other than normal operation.

**Appendix A – Mean and Standard Deviation of Elementary Effects**

To identify which parameters are the most sensitive, some researchers compare the average of the EE values for the different parameters across all input starting points. Additionally, some look at the standard deviation of the EE values for a given parameter across the different starting points. This helps to identify large sensitivity variation at different points, indicating strong interaction with the values of other parameters. As commonly found in EE-related literature, EE analysis typically identifies the most sensitive parameters using a plot to pictorially show the standard deviation versus mean values of the EE values. However, it is difficult to systematically identify the most sensitive parameters using this approach.

The mean of the absolute EE value for the ultimate loads for output $o$, input parameter $i$, and bin $b$ is calculated as:

$$\mu^*_{oib} = \frac{1}{R} \sum_{r=1}^{R} |EE^r_{oib}| \tag{11}$$

where $R$ is the number of points at which the EE value is calculated. The standard deviation of the EE is then calculated as:

$$\sigma_{oib} = \sqrt{\frac{1}{(R-1)} \sum_{r=1}^{R} \left(EE^r_{oib} - \mu_{oib}\right)^2} \tag{12}$$

and $\mu_{oib}$ is defined as:

$$\mu_{oib} = \frac{1}{R} \sum_{r=1}^{R} EE^r_{oib} \tag{13}$$

This is shown in Figure 24 and Figure 25 for the blade-root bending ultimate moment and the blade-root bending OoP fatigue moment metrics for both the wind parameter and turbine parameter case studies, respectively. Shown in Figure 24 is the large 20 sensitivity of shear in the lowest wind speed bin and the large sensitivity of the u-turbulence across all wind speed bins. Shown in Figure 25 is the large sensitivity of yaw error in the below-rated wind speed bin and the large sensitivity of the lift distribution at the outboard section of the blade in the below- and near-rated wind speed bins.




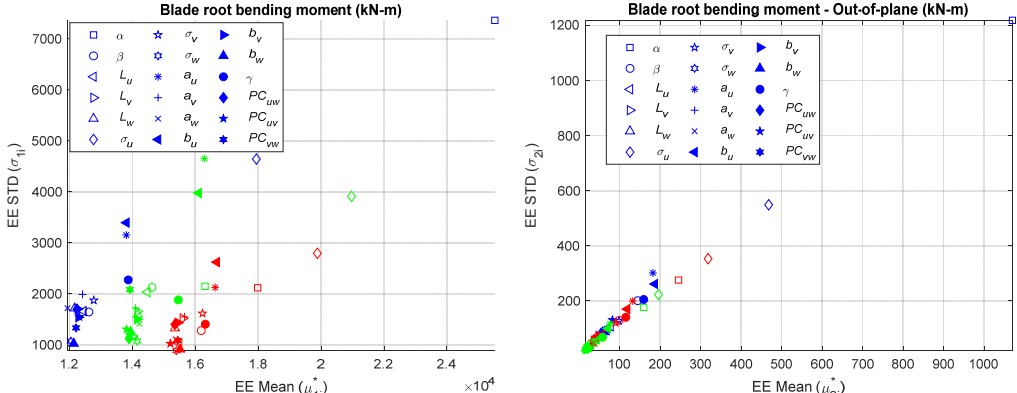

**Figure 24: EE standard deviation vs EE mean for blade-root bending moment ultimate load (left) and blade-root out-of-plane bending moment fatigue load (right) at all wind speed bins (blue=below rated, red=near rated, green=above rated).**

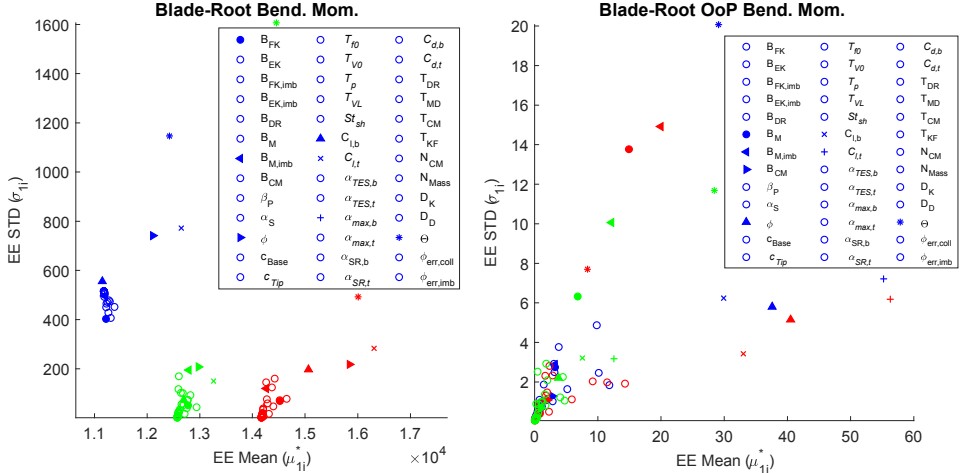

**Figure 25: EE standard deviation vs EE mean for blade-root bending moment ultimate load (left) and blade-root out-of-plane bending moment fatigue load (right) at all wind speed bins (blue=below rated, red=near rated, green=above rated).**

### Acknowledgements

This work was authored by Alliance for Sustainable Energy, LLC, the manager and operator of the National Renewable Energy
Laboratory for the U.S. Department of Energy (DOE) under Contract No. DE-AC36-08GO28308. Funding provided by
Department of Energy Office of Energy Efficiency and Renewable Energy, Wind Energy Technologies Office. The views
expressed in the article do not necessarily represent the views of the DOE or the U.S. Government. The U.S. Government
retains and the publisher, by accepting the article for publication, acknowledges that the U.S. Government retains a
nonexclusive, paid-up, irrevocable, worldwide license to publish or reproduce the published form of this work, or allow others
to do so, for U.S. Government purposes.





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
