# Peer review of "Sensitivity Analysis of Wind Characteristics and Turbine Properties on Wind Turbine Loads"

_Wind Energy Science, 2019_

## Referee Comment (RC1) · Imad Abdallah (Referee) · 24 Feb 2019

//————————————————-// General comments //————————————————-//

The date of Reference number 2 ("Assessment of extreme design loads for modern wind turbines using the probabilistic approach," DTU Wind Energy (DTU Wind Energy PhD; No. 0048(EN)) should be 2015 and not 2018.

Early in the paper, the authors should consider explaining their logic for choosing to use the Elementary Effects sensitivity approach instead of other approaches. As far as I am concerned EE sensitivity type of analysis is mainly used for initial assessments of input

parameters, when you have large number of input parameters and it only provides information in the qualitative sense: indicates influential vs non-influential input, and hints to higher order effects caused by nonlinear or interactive relationship between parameters. You briefly explain this in section 3.1, but maybe you should consider summarizing the logic in your intro.

//——————————————-// Specific comments //——————————————-//

Page 2, Lines 4-6: I don't fully agree. Say we have a long and slender blade. You use ElastoDyn for the to define the blade dynamics via 1-2 assumed flap and 1-edge modes. This means that all your structural dynamics are effectively filtered through those three modes. A complex combination of wind speed, turbulence, shear, veer and yaw error might -in reality- result in a bend-twist coupling that will increase the loads or, unintuitively, reduce the loads (because the twist results in lower angles of attack). You will never be able to capture such a phenomena with a simpler model resulting in erroneous conclusions on your sensitivity analysis.

Page 2, Lines 13-14: What do you see the downfall of your sensitivity analysis if correlations and joint distributions of input parameters are not taken into account? See for instance slides 22 and 23 in http://www.gdr-mascotnum.fr/media/mascot12caniou.pdf On the same topic, since you use ranges, you might easily fall into an erroneous case where high wind speeds and large shear exponents (alfa>1) combine. . .but I could see from Table 3 that you chose your ranges and combinations carefully (you also make this clear on lines 4.8, page 10).

Page 3, Lines 23-24: could you please explain the reason for choosing the vector sum of the components of the bending moments? Imagine bending moment Mx is an order of magnitude larger than bending moment My. Under some combination of the input, we observe that My exhibit large variations (x2 or x3) where as Mx doesn't. However, given that Mx is an order of magnitude larger than My, the large fluctuations of My will not be really reflected in the vector sum. Consequently, the sensitivity analysis will not
reflect the real effects of the input any longer.

Page 10, lines 4-6: I actually propose you compare the sensitivity analysis performed on the same set of input assuming they are independent and then assuming joint distributions (where possible).

Despite my previous comment, your results in Figures 3 & 4 and your summary on page 18, conform to results found by investigators/researchers. So, I don't think you have introduced flagrant errors using the approach proposed in this article.

Section 4.2.2.4 Steady Airfoil Aerodynamics - Abdallah et al. proposed the initial probabilistic model. You made some nice modifications and contributions. I propose we make both models available to the public (open source, open access), in order for further future improvements be made by others. - Figure 13 shows samples of perturbed Cl and Cd curves. I notice that the Cl perturbations for positive angles of attack are shown but not for the negative angles of attack. Does this mean that the model does not handle Cl perturbations for negative angles of attack? If not it should, especially that you consider ultimate loads and large yaw errors. - It is not clear if you maintain the correlations of Cl and Cd curves along the span of the blade?

Control properties, Table 10 - -20 to 20 degrees is a fairly large range for standard yaw error for a turbine in normal operation and connected to the grid, which might explain why this parameter ends up being so significant as shown in Figure 13, 14, and Table 11. Unless the underlying assumption is that this range implicitly includes the effect of rapid directional change of the wind. In principle a controller should be able to detect such large yaw errors (say over a 30-60 second averaging windows) and perform the necessary safety procedure (whatever that might be).

Figure 13 has the grid on, Figure 14 has the grid off.

Page 25, Line 13: "Ultimate turbine loads are most sensitive to yaw error (theta) and lift (Cl) distribution" I would say "Both Ultimate and fatigue loads are…"

Page 27, Lines 3-17: very good discussion. Useful information here that needs to be carried out to future investigations!

Page 35, Line 8-10: very good discussion. Useful information here that needs to be carried out to future investigations!

---

## Referee Comment (RC2) · Anonymous Referee #2 · 4 Mar 2019

**1 General comments**

**1.1 Summary of key points**

The paper is well written and its topic is relevant. The goals are clearly stated: sensitivity analyses for the NREL 5 MW turbine. They are ambitious because a large number of input and output variables are involved and a computationally demanding model (OpenFAST-based) is used. The choice to reduce the complexity of the analysis by using the relatively simple Elementary Effects approach is judicious.

[Figure]

However, it seems to me that nevertheless, the problem is still too complex. To be able to tackle it, the authors work with a relatively small set of input vectors. This is the main weakness I find to be present in their analysis and they have not convincingly argued that the number of input vectors is sufficient.

A further issue is their adaptation of the Elementary Effects approach in a way that is insufficiently justified. The current exposition leaves me doubting that it is really a consistent sensitivity analysis approach. This does not mean that all their conclusions are arbitrary. On the contrary, I would guess that many conclusions about sensitivities are correct due to their broadly consistent nature over the whole input space and remain unaffected by details of the sensitivity analysis. (This may mean that they would also appear in simpler analyses and could perhaps be obtained from expert elicitation.)

Despite my rather negative judgment about the method and assumptions, there are some gems in this paper. Notably, the authors' efforts in obtaining useful ranges for input variables have resulted in overview tables that are more broadly useful in their own right.

**1.2 Overview of review aspects**

My judgments here are based on my current understanding of the work. Brief justifications are given, but detail and nuance for the negative comments are postponed to the 'Specific comments' section. They may change due to clarification by author comments.

1. *Does the paper address relevant scientific questions within the scope of WES?*
   **Yes.** Knowledge about sensitivities is of great value to wind turbine design and selection.

2. *Does the paper present novel concepts, ideas, tools, or data?*

**Some.** Novel variants of Elementary Effects sensitivity analysis are presented. A lot of interesting, new simulation data was generated and used.

3. *Is the paper of broad international interest?*
   **Yes.** The discussion is relevant for all locations and in various wind energy research subdomains.

4. *Are clear objectives and/or hypotheses put forward?*
   **Yes.** To provide sensitivities of relevant output variables relative to coherent sets of input variables.

5. *Are the scientific methods valid and clear outlined to be reproduced?*
   **Validity may be tenuous and reproduction would be difficult.** (i) I have my doubts that the novel variants of Elementary Effects sensitivity analysis is a proper sensitivity analysis. (ii) The determination of input variables is not discussed in sufficient detail for them to be even approximatively recreated.

6. *Are analyses and assumptions valid?*
   **One important one is not.** I find it doubtful that the set of input vectors is large enough to warrant conclusions as concrete as the ones drawn, even more so given that a nontrivial number of them may not correspond to physical situations.

7. *Are the presented results sufficient to support the interpretations and associated discussion?*
   **No.** This is a consequence of my evaluation of the two preceding points.

8. *Is the discussion relevant and backed up?*
   **Yes.** Based on the results the authors present, the conclusions are reasonable.

9. *Are accurate conclusions reached based on the presented results and discussion?*
   **Ambiguous.** Given the presented results and discussion, the conclusions are

accurate, but I think that flaws in the analysis and assumptions cast doubt on that accuracy.

10. *Do the authors give proper credit to related and relevant work and clearly indicate their own original contribution?*
**Yes.** According to my knowledge they certainly do. And notably, they make very good use of information in the literature for ranges of input variable values.

11. *Does the title clearly reflect the contents of the paper and is it informative?*
**It can be improved.** Currently, the title implies a scope that is larger than in actuality and is a bit long and complex. Suggestion: "Elementary effects sensitivity analyses of the NREL 5 MW turbine"

12. *Does the abstract provide a concise and complete summary, including quantitative results?*
**Yes, but.** (i) Quantitative results are not given, but neither are they appropriate. (ii) The future applications listed are not sufficiently discussed in the paper to be included.

13. *Is the overall presentation well structured?*
**Yes.**

14. *Is the paper written concisely and to the point?*
**Yes.** There is a bit of repetition in the presentation of the second case study, but this redundance may actually make the paper easier to read.

15. *Is the language fluent, precise, and grammatically correct?*
**Yes.**

16. *Are the figures and tables useful and all necessary?*
**Not all.** I found Figures 5–8, 10–11, and 15–22 of limited usefulness; the information should be more filtered and summarized to be useful. In contrast to these stand especially Figures 9 and 23.

17. *Are mathematical formulae, symbols, abbreviations, and units correctly defined and used according to the author guidelines?*
**Not all or in all aspects.** Standards about notation of variables and constants are not followed and a mixture of fonts is distractingly used for mathematical notation.

18. *Should any parts of the paper (text, formulae, figures, tables) be clarified, re-duced, combined, or eliminated?*
**Yes.** For example, Figures 5–8, 10–11, and 15–22 as mentioned above.

19. *Are the number and quality of references appropriate?*
**Yes.**

20. Is the amount and quality of supplementary material appropriate and of added value?
**No supplementary material has been provided.** It would have been useful if the simulation data (inputs, outputs) were made avaiable, but it is of course the prerogative of the authors not to do so.

**2  Specific comments**

**2.1  Your modified EE formulae**

There is insufficient justification of your modified formulae.

You indicate why you add $\bar{Y}_{ob}$ in Eq. (3) and use a dimensional version, but just men-tioning it is insufficient as an explanation. To me, adding a constant term to a set of

sensitivities will substantially distort the information present in the quantity; it is not a sensitivity anymore. I can sense a reason for making it dimensional, but I can think of other reasons why this is a bad idea; you should pre-emptively remove such doubts. Currently, I am not convinced at all that what you call a sensitivity here can be interpreted and used as such.

In Eq. (5), you multiply by a probability and again indicate why, but do not explain it at all. Here, I can guess at the reason.

Part of my reticence here is due to the fact that I am skeptical of your approach to the identification of most sensitive inputs; this is discussed next. As your modifications here are, I assume, related to your non-standard approach, it may be good to explain them concurrently in the text.

**2.2 Your approach to the identification of most sensitive inputs**

First of all, by looking at plain means, you implicitly assume that your samples are uniformly distributed over the input space. Even if you cannot justify this, you should at least discuss the implications. Related to this, you apparently do not calculate the expectation over wind speeds. Of course, for fatigue loads this is actually what happens because you have included the probability in the EE value. For ultimate loads, it does not, where I see no reason why you should throw away the probability information that you do have here and replace it with a uniform distribution (implicit by taking the plain mean, as said before).

Second, you say that you do not use the standard approach, but do not really justify that. You refer to an appendix and there is information about that there, but take into account that appendices are meant for skippable material. When reading your approach and the standard one as sketched in the appendix, the latter was more appealing to me. The reason is that there the rankings are primarily based on the means.

You base your rankings essentially on tail behavior of the EE value distribution. Tail behavior says very little about what happens in the bulk of the distribution. It may well be that a distribution with a low mean but a fat tail generates more instances above your somewhat arbitrarily defined threshold than a distribution with a high mean but a skinny tail. Based on this reasoning, I fear that your approach may unduly rank higher inputs that generate fat tails, based just on the tail and not on the mean. It is possible that the tail behavior is similar over the inputs and then your criterion will work, but you do not show or discuss that. Even in that case I see no reason not to focus more on the means. So, a good start to convincing me is to explain why a ranking of means is not appropriate here and how your criterion overcomes the problems apparently present in other approaches.

**2.3 Your selection of input vectors**

You use thirty input vectors times three—one for each wind speed bin—for both case studies. The quality of the selection of these points is essential, but the way in which you choose them is dealt with only in a single sentence where you claim that by using Sobol numbers—without providing details—you can ensure a wide sampling of the input hyperspace. Furthermore, you choose to ignore the dependencies between the input variables, but directly sample from the Cartesian product of the individual ranges you have defined, so non-physical input vectors could be included. Finally, your input spaces are very large, 18-dimensional and 40-dimensional respectively.

In such high-dimensional spaces, even a few dependencies can cause the subset of physical vectors to be 'small' relative to the Cartesian product. So it may be difficult to actually land on a physical vector, when those dependencies are not taken into account. (I do not see how using Sobol numbers can help with this issue, as more even sampling cannot correct for ignored dependencies.) This may mean that *I cannot exclude the possibility that the majority or even all of the input vectors you use is non-*

*physical*. The fact that 90 input vectors in such high-dimensional spaces is a very small sample, only makes this issue more problematic. (I understand that for computational time reasons you cannot increase this number by orders of magnitude.) Because a priori I must assume that non-physical input vectors may result in non-realistic output sensitivities, this issue undermines your results.

In the conclusion, you state that "The combinations of parameters in this study spanned the ranges of several different locations." Given the reasoning I developed above, you will understand that I am skeptical of this. But this can be tested: how representative are your input vectors of existing locations? This may provide an avenue to reduce my worries about your selection of input vectors.

**2.4  Your presentation of applications and future work**

In the conclusions, you present a number of possible applications of your work and also ideas for future work. I feel that some—about error bars and insight—are described to briefly or too vague to really know whether indeed, your results provide a useful starting point.

**2.5  Compliments**

**p.2, §2.1, list**  Nice and clear overview of limitations.

**p.4, §3.1**  Nice, compact overview.

**p.4, §3.2**  Well-explained.

**p.10, Table 3**  Very useful overview table.

**p.18, text; p.19 Tables 4–5**  This (type of) summarizing is very useful.

**3  Technical corrections**

**3.1  General**

- In mathematical notation, the following standard conventions are prescribed: variables are written in a cursive/italic/slanted font and constants are written in a roman/upright font.

```
https://www.nist.gov/pml/nist-guide-si-chapter-10-more-
    printing-and-using-symbols-and-numbers-scientific
                   -and-technical#102
```

Please do so throughout, as currently this is not adhered to and most every symbol is put in italic font, even word abbreviations (which are certainly constant).

- As per the SI standards, between a value and a unit there should always be a normal (unbreakable) space. So do not write '5-MW', but '5 MW'; there are other examples in the text where no space is present. Also, composed units should not be separated by a hyphen, use a centered dot instead; e.g., 'kN·m' instead of 'kN-m'.

- You abuse the same symbol for functions and variables: for example $Y$ and $Y_o$ instead of $f$ and $f_o$; avoid that, as it causes confusion.

- Your presentation of the EE approach is symbolically far more verbose than it needs to be. Suggestions:

  - Make effective use of vectorial notation. For example, Eq. (2) could be written as

$$EE_i = \frac{f(\vec{U} + \Delta \vec{e}_i) - f(\vec{U})}{\Delta}.$$

(Arrows instead of bold here only because of reviewing system limitations.)

– Use consistent notation for elements (lower case) and sets (upper case). For example $u \in U$, $y$ instead of $Y$.

– Make subscripts more 'direct' by avoiding index layers and using variables directly. For example, $v$ instead of both $b$ and $v_b$, $u$ instead of both $r$ and $u^r$.

– Put commas between subscripts and do not overburden subscript/superscript locations.

– Predefine some recurring fragments, such as $\Delta_i^v = \max U_i^v - \min U_i^v$ and $\delta_i^v = \Delta_i^v/10$.

Combined, you could write Eq. (3), (4), and (5) as

$$EE_{i,o}^v(\vec{u}^v) = \bar{y}_o^v + 10 \left| f_o(\vec{u}^v \pm \delta_i^v \vec{e}_i) - f_o(\vec{u}^v) \right| \quad \text{with} \quad f_o(\vec{u}^v) = \frac{1}{|S|} \sum_{s \in S} \max_t |f_{o,s}(\vec{u}^v, t)|,$$

$$EE_{i,o}^v(\vec{u}^v) = 10 \left| g_o(\vec{u}^v \pm \delta_i^v \vec{e}_i) - g_o(\vec{u}^v) \right| p(v) \quad \text{with} \quad g_o(\vec{u}^v) = ...$$

• You use 'parameters' where I would use 'variables', given that the focus is on *varying* those quantities. For example, in your setup, I would call wind speed a variable, as it parameterizes some sub-cases, but is not varied as, e.g, wind shear is.

• Throughout the paper, the formulation of sensitivities are according to my ear often reversed. In the paper, input variables are called '(most) sensitive', whereas I would apply such language to output variables only. For input variables, words like 'impactful' and 'influencing' come to mind, although these do not seem ideal. Consider changing the language, but for the input side feel free to find a better word than the ones I came up with.

• Do not break tables over multiple pages; if you must, repeat the header row.

**3.2 Local**

**p.2, l.33** OpenFAST is software, not an approach.

**p.3, l.6** What does 'down-selected' mean?

**p.3, l.13** You *must* say how this was assessed.

**p.4, Table 1** This nice table can be improved by adding symbols and a clear indication of which variables are used for which case study (wind vs. turbine and ultimate vs. fatigue).

**p.4, §3.1** Paragraph way too long; split in two or three.

**p.5, l.20** Elaborate, the current explanation about the 'radial approach' is too limited.

**p.5, l.22** Elaborate, the current explanation is too limited. How did you obtain the Sobol numbers and how did you use them.

**p.6, l.1** Say how and why it was modified or at least reference forward.

**p.6, l.8** 'nondimensionalizing' to 'making dimensionless/adimensional'

**p.6, l.9** 'derivative' to 'finite difference'

**p.6, l.10** 'IEC turbine class $I$ and category $B$' to 'IEC turbine class I and category B' (to avoid confusion with index sets $I$ and $B$)

**p.6, Eq. (3) and (5)** $ib \pm 1$ to $bi \pm 1$ (but better still follow my general suggestions regarding math notation to avoid making such mistakes)

**p.6, Eq. (4)** In principle, the LHS depends on the set of seeds

**p.6, l.20** Why choose the sign randomly? Does this matter?
**p.6, l.20 and elsewhere** 'IEC-Class IB' to 'IEC Class IB'

**p.6, l.20** move 'The added term . . . ' forward in section.

**p.6, l.23** Define DEL (as you should all letter words); do not assume all your readers will be familiar with this.

**p.7, l.6** Your notation includes indices that are in fact averaged away ($r$, $i$, $b$) for the mean EE value, but for the (sample) standard deviation, you include not even the one that isn't gone ($o$). Some consistent, standard notation would be appropriate here; consider something like the sample mean $m_o$ and sample standard deviation $s_o$, perhaps.

**p.7, l.8** The part about 'stratification' requires more explanation, certainly because it is not immediately clear in what way this differs between the two case studies. (Perhaps you can reference to a relevant earlier part of the text.)

**p.7, l.8–12** The way in which you decide to include or exclude some wind speed bins comes accross as somewhat arbitrary here. More explanation may be needed.

**p.7, l.17** 'speed bins' to 'speeds'

**p.7, l.24** 'Holtstag' to 'Holtslag'

**p.7, l.35** 'either better optimized or lower risk': optimized for what? lower risk of what? Avoid vagueness here, as concrete applications are a good justification for your work.

**p.8, Table 2** use the same, correct mathematical symbol, e.g., 'u ($\sigma_u$)'.

**p.8, l.16** Suggestion: for $q$, use the velocity component name directly, instead of an index.

**p.8, l.19–21** Giving the exact quantity definitions of the IEC standard here is superfluous, certainly because you do not use it.

**p.8, l.22** 'and random' to 'and to the random' (?)

**p.9, l.9–10** 'term' to 'factor'

**p.9, l.11** 'in (IEC, 2005)' to 'in the standard (IEC, 2005)' (text should make sense when reading aloud, skipping over citation parentheticals)

**p.9, l30** '$B$' to 'B'

**p.10, Table 3 header**   • There is no need for parentheses around units.
   • $b_u$, $b_v$, $b_w$ are not dimensionless; I suggest using 1/mm for convenient notation of the values.

**p.10, Table 3 footnote** Elaborate a bit.

**p.10, l.7** 'unphysical' to 'non-physical'

**p.11, Fig. 3–4** Suggestion: use a logarithmic axis for the counts, then zooming will not be necessary and more information should become visible.

**p.19, Tables 4–5** Make it explicit in the table whether the numbers given are percentages or counts.

**p.20, l.8–9** Provide more information about the expert(s) and the elicitation procedure used.

**p.20, Table 6**   • Do *not* use the empty set symbol $\emptyset$ instead of the Greek letter phi $\phi$.
   • $B_{M,imb}$ to $B_{M,imb}$ (just an example of the type of math font use corrections that you should implement; note that the font of the subscripts, which here refer to words, are upright)

**p.21–22, list around page break**  What are the parenthetical, bold math symbols for?

**p.22, Figure 12**  Avoid putting figures in the middle of paragraphs.

**p.23, l.18–19**  Avoid line breaking tuples.

**p.23, l.19–20**  Use paragraph breaks as defined by the style.

**p.24, Figure 13**  • Are curves for $C_{\text{d,orig}}$ and $C_{\text{d},-10\%}$ missing here, or do they just overlap?

   • This Figure is not referenced in the text, I think. (All figures and tables should be.)

**p.25, l.8–23**  This paragraph is too long; split.

**p.25, l.17**  'relevant' to 'relative' (?)

**p.27, l.3–17**  This paragraph is too long; split.

**p.37, l 1**  'hear' to 'here'

**p.38, Figures 24–25**  I think that plots of $\sigma/\mu$ vs. $\mu$ would provide more insight here.

---

## Referee Comment (RC3) · Anonymous Referee #3 · 4 Mar 2019

- Very interesting paper with very interesting and useful results. - The baseline methodology should be described in more detail in order to stand out as reference. For other wind systems. In particular sections 3.3.1 and 3.3.2 should be revised to clarify to the reader in which way the initial function was adjusted. This is done good on a detailed level, but it would help to get a higher-level summary of the idea behind the applied procedure. Also, it should be highlighted throughout the paper that the approach is a tailored approach for the problem at hand. The alteration of the baseline formulas as well as the threshold evaluation of EE indicates that classical SA is not applied. There are good reasons presented for it, and i expect the results to be valid nonetheless, but the variation of standard approaches is significant. - The plots are

quite comprehensive. It could be sufficient to just show exemplary plots to describe the applied methodology, and provide the full set of plots inside the appendix. The summarizing tables are very helpful. - EE is generally used as screening method, in order to identify relevant input parameters. In this sense, the values of the resulting EE should be handled with care. They only provide an indicator of relevance, not of the sensitivity (or even the comparative relevance). This should be taken into account when evaluating the resulting EEs.

Please also note the supplement to this comment:
https://www.wind-energ-sci-discuss.net/wes-2019-2/wes-2019-2-RC3-supplement.pdf

**Supplement:**

[revised manuscript text omitted]

---

## Author Comment (AC1) · 15 Apr 2019

The authors thank the reviewer for his/her thorough assessment, comments, and insights. Revised text addressing these comments is shown in the new version of the manuscript as highlighted in red.

The reviewer's comments have been copied in the attached document, and are shown in blue. Authors' responses are shown in black.

Please also note the supplement to this comment:
https://www.wind-energ-sci-discuss.net/wes-2019-2/wes-2019-2-AC1-supplement.pdf

[Figure]

**Authors' Note to the Associate Editor and Reviewers**

**Title: Sensitivity of Uncertainty in Wind Characteristics and Wind Turbine Properties on Wind Turbine Extreme and Fatigue Loads**

**Ref. No: wes-2019-2**

The authors thank the reviewers for their thorough assessment, comments, and insights. Revised text in the manuscript is highlighted in red.

Reviewer's comments are shown in blue. Authors' responses are shown in black.

The date of Reference number 2 ("Assessment of extreme design loads for modern wind turbines using the probabilistic approach," DTU Wind Energy (DTU Wind Energy PhD; No. 0048(EN)) should be 2015 and not 2018.
A.      This has been corrected in the reference itself and any mention of the reference throughout the paper.

Early in the paper, the authors should consider explaining their logic for choosing to use the Elementary Effects sensitivity approach instead of other approaches. As far as I am concerned EE sensitivity type of analysis is mainly used for initial assessments of input parameters, when you have large number of input parameters and it only provides information in the qualitative sense: indicates influential vs non-influential input, and hints to higher order effects caused by nonlinear or interactive relationship between parameters. You briefly explain this in section 3.1, but maybe you should consider summarizing the logic in your intro.
A.      We have added some new information to the introduction based on this advice.

Page 2, Lines 4-6: I don't fully agree. Say we have a long and slender blade. You use ElastoDyn for the to define the blade dynamics via 1-2 assumed flap and 1-edge modes. This means that all your structural dynamics are effectively filtered through those three modes. A complex combination of wind speed, turbulence, shear, veer and yaw error might -in reality- result in a bend-twist coupling that will increase the loads or, unintuitively, reduce the loads (because the twist results in lower angles of attack). You will never be able to capture such a phenomena with a simpler model resulting in erroneous conclusions on your sensitivity analysis.
A.       It has been shown in many studies that ElastoDyn is sufficiently accurate for loads analysis of the NREL 5 MW turbine blade. This has been shown via code-to-code comparisons to BeamDyn, MSC.ADAMS, HAWC2, Bladed, etc.

Page 2, Lines 13-14: What do you see the downfall of your sensitivity analysis if correlations and joint distributions of input parameters are not taken into account? See for instance slides 22 and 23 in http://www.gdr-mascotnum.fr/media/mascot12caniou.pdf
A.      Correlations and joint distributions of the parameters were not considered since developing this relationship for so many parameters would be difficult or impossible.  In addition, the correlation could be very different for different wind sites.  The impact of not considering the correlation was limited by choosing parameters that were fairly independent of one another, when possible, and by binning the results by wind speed.

**Fig. 1.**

**Supplement:**

**Authors' Note to the Associate Editor and Reviewers**

**Title: Sensitivity of Uncertainty in Wind Characteristics and Wind Turbine Properties on Wind Turbine Extreme and Fatigue Loads**

**Ref. No: wes-2019-2**

The authors thank the reviewers for their thorough assessment, comments, and insights. Revised text in the manuscript is highlighted in red.

Reviewer's comments are shown in blue. Authors' responses are shown in black.

The date of Reference number 2 ("Assessment of extreme design loads for modern wind turbines using the probabilistic approach," DTU Wind Energy (DTU Wind Energy PhD; No. 0048(EN)) should be 2015 and not 2018.

A.        This has been corrected in the reference itself and any mention of the reference throughout the paper.

Early in the paper, the authors should consider explaining their logic for choosing to use the Elementary Effects sensitivity approach instead of other approaches. As far as I am concerned EE sensitivity type of analysis is mainly used for initial assessments of input parameters, when you have large number of input parameters and it only provides information in the qualitative sense: indicates influential vs non-influential input, and hints to higher order effects caused by nonlinear or interactive relationship between parameters. You briefly explain this in section 3.1, but maybe you should consider summarizing the logic in your intro.

A.        We have added some new information to the introduction based on this advice.

Page 2, Lines 4-6: I don't fully agree. Say we have a long and slender blade. You use ElastoDyn for the to define the blade dynamics via 1-2 assumed flap and 1-edge modes. This means that all your structural dynamics are effectively filtered through those three modes. A complex combination of wind speed, turbulence, shear, veer and yaw error might -in reality- result in a bend-twist coupling that will increase the loads or, unintuitively, reduce the loads (because the twist results in lower angles of attack). You will never be able to capture such a phenomena with a simpler model resulting in erroneous conclusions on your sensitivity analysis.

A.        It has been shown in many studies that ElastoDyn is sufficiently accurate for loads analysis of the NREL 5 MW turbine blade. This has been shown via code-to-code comparisons to BeamDyn, MSC.ADAMS, HAWC2, Bladed, etc.

Page 2, Lines 13-14: What do you see the downfall of your sensitivity analysis if correlations and joint distributions of input parameters are not taken into account? See for instance slides 22 and 23 in http://www.gdr-mascotnum.fr/media/mascot12caniou.pdf

A.        Correlations and joint distributions of the parameters were not considered since developing this relationship for so many parameters would be difficult or impossible.  In addition, the correlation could be very different for different wind sites.  The impact of not considering the correlation was limited by choosing parameters that were fairly independent of one another, when possible, and by binning the results by wind speed.

On the same topic, since you use ranges, you might easily fall into an erroneous case where high wind speeds and large shear exponents (alfa>1) combine ... but I could see from Table 3 that you chose your ranges and combinations carefully (you also make this clear on lines 4.8, page 10).

A.      The authors agree and correlations on inflow parameters were minimized by binning the results by mean wind speed.

Page 3, Lines 23-24: could you please explain the reason for choosing the vector sum of the components of the bending moments? Imagine bending moment Mx is an order of magnitude larger than bending moment My. Under some combination of the input, we observe that My exhibit large variations (x2 or x3) whereas Mx doesn't. However, given that Mx is an order of magnitude larger than My, the large fluctuations of My will not be really reflected in the vector sum. Consequently, the sensitivity analysis will not reflect the real effects of the input any longer.

A.      It is common practice in axisymmetric structure responses (blade root, drivetrain, tower) to only consider the vector magnitude of the bending moments in ultimate loads analysis.

Page 10, lines 4-6: I actually propose you compare the sensitivity analysis performed on the same set of input assuming they are independent and then assuming joint distributions (where possible).

A.      It would not be possible to make an 18-dimensional joint probability distribution. The purpose of this study is to take a very large set of input parameters and identify the parameters that most contribute to turbine response sensitivity. As such, including 18-dimensional joint distributions is not possible and was not considered for the present study. However, it would be beneficial to include joint probability distributions in future studies that include fewer parameters.

Despite my previous comment, your results in Figures 3 & 4 and your summary on page 18, conform to results found by investigators/researchers. So, I don't think you have introduced flagrant errors using the approach proposed in this article.

A.      Agreed.

Section 4.2.2.4 Steady Airfoil Aerodynamics - Abdallah et al. proposed the initial probabilistic model. You made some nice modifications and contributions. I propose we make both models available to the public (open source, open access), in order for further future improvements be made by others.

A.      The authors agree that this would be beneficial to the research community.

Figure 13 shows samples of perturbed Cl and Cd curves. I notice that the Cl perturbations for positive angles of attack are shown but not for the negative angles of attack. Does this mean that the model does not handle Cl perturbations for negative angles of attack? If not it should, especially that you consider ultimate loads and large yaw errors.

A.      Analysis was intentionally limited to only consider normal range of operation. As such, the Cl perturbations were limited to the range between the beginning of the linear Cl region and 90°. The beginning of the linear region is found to begin as low as $\alpha = -13°$.

It is not clear if you maintain the correlations of Cl and Cd curves along the span of the blade?

A. The perturbations are made at the blade tip and root and interpolated to the airfoil data between these extremes.

Control properties, Table 10 - -20 to 20 degrees is a fairly large range for standard yaw error for a turbine in normal operation and connected to the grid, which might explain why this parameter ends up being so significant as shown in Figure 13, 14, and Table 11. Unless the underlying assumption is that this range implicitly includes the effect of rapid directional change of the wind. In principle a controller should be able to detect such large yaw errors (say over a 30-60 second averaging windows) and perform the necessary safety procedure (whatever that might be).

A. This is the value found in the literature and seems reasonable given the authors' discussions with experts.

Figure 13 has the grid on, Figure 14 has the grid off.

A. This has been fixed.

Page 25, Line 13: "Ultimate turbine loads are most sensitive to yaw error (theta) and lift (Cl) distribution" I would say "Both Ultimate and fatigue loads are..."

A. As fatigue loads are not most sensitive to the lift distribution, this adjusted statement would be incorrect.

Page 27, Lines 3-17: very good discussion. Useful information here that needs to be carried out to future investigations!

A. Agreed.

Page 35, Line 8-10: very good discussion. Useful information here that needs to be carried out to future investigations!

A. Agreed.

---

## Author Comment (AC3) · 15 Apr 2019

The authors thank the reviewers for their thorough assessment, comments, and insights. Revised text in the attached revised manuscript is highlighted in red.

Reviewer's comments have been copied into the attached document and are shown in blue. Authors' responses are shown in black.

Please also note the supplement to this comment:
https://www.wind-energ-sci-discuss.net/wes-2019-2/wes-2019-2-AC3-supplement.pdf

[Figure]

[Figure]

**Authors' Note to the Associate Editor and Reviewers**

**Title: Sensitivity of Uncertainty in Wind Characteristics and Wind Turbine Properties on Wind Turbine Extreme and Fatigue Loads**

**Ref. No: wes-2019-2**

The authors thank the reviewers for their thorough assessment, comments, and insights. Revised text in the manuscript is marked in red.

Reviewer's comments are shown in blue. Authors' responses are shown in black.

1 General comments
1.1 Summary of key points
The paper is well written and its topic is relevant. The goals are clearly stated: sensitivity analyses for the NREL 5 MW turbine. They are ambitious because a large number of input and output variables are involved and a computationally demanding model (OpenFAST-based) is used. The choice to reduce the complexity of the analysis by using the relatively simple Elementary Effects approach is judicious. However, it seems to me that nevertheless, the problem is still too complex. To be able to tackle it, the authors work with a relatively small set of input vectors. This is the main weakness I find to be present in their analysis and they have not convincingly argued that the number of input vectors is sufficient.
A.      The authors parameterized inflow inputs based on the capabilities of TurbSim and parameterized the turbine inputs to represent the main physical effects where uncertainties were probable.

A further issue is their adaptation of the Elementary Effects approach in a way that is insufficiently justified. The current exposition leaves me doubting that it is really a consistent sensitivity analysis approach. This does not mean that all their conclusions are arbitrary. On the contrary, I would guess that many conclusions about sensitivities are correct due to their broadly consistent nature over the whole input space and remain unaffected by details of the sensitivity analysis. (This may mean that they would also appear in simpler analyses and could perhaps be obtained from expert elicitation.)
A.      Elementary Effects at its fundamental level can only be considered a screening method. However, the introduction of the use of Sobol numbers and radial trajectories increases its efficacy as a method for estimating sensitivity, not just as a screening method.  Campolongo empirically demonstrated that the results obtained by EE can converge to a variance-based sensitivity index with increased number of Sobol points.  In this work, the authors increased the number of Sobol starting points until the EE-based sensitivity metrics had shown convergence.

Despite my rather negative judgment about the method and assumptions, there are some gems in this paper. Notably, the authors' efforts in obtaining useful ranges for input variables have resulted in overview tables that are more broadly useful in their own right.
A.      Agreed.

**Fig. 1.**

---

## Author Comment (AC4) · 15 Apr 2019

**Authors' Note to the Associate Editor and Reviewers**

**Title: Sensitivity of Uncertainty in Wind Characteristics and Wind Turbine Properties on Wind Turbine Extreme and Fatigue Loads**

**Ref. No: wes-2019-2**

The authors thank the reviewers for their thorough assessment, comments, and insights. Revised text in the manuscript is highlighted in red.

Reviewer's comments are shown in blue. Authors' responses are shown in black.

Very interesting paper with very interesting and useful results. The baseline methodology should be described in more detail in order to stand out as reference. For other wind systems. In particular sections 3.3.1 and 3.3.2 should be revised to clarify to the reader in which way the initial function was adjusted. This is done good on a detailed level, but it would help to get a higher-level summary of the idea behind the applied procedure.

A.       $Y_b$ was necessary to properly compare ultimate load consistently across the bins because it is only the maximum ultimate load that matters. The scaling by the probability was necessary to properly compare fatigue loads consistently across the bins because it is the cumulative effect of all bins that matters for fatigue.

Also, it should be highlighted throughout the paper that the approach is a tailored approach for the problem at hand. The alteration of the baseline formulas as well as the threshold evaluation of EE indicates that classical SA is not applied. There are good reasons presented for it, and i expect the results to be valid nonetheless, but the variation of standard approaches is significant.

A.       Some language was added to the paper to address this comment.

The plots are quite comprehensive. It could be sufficient to just show exemplary plots to describe the applied methodology, and provide the full set of plots inside the appendix. The summarizing tables are very helpful.

A.       Some plots have been moved to the appendix.

EE is generally used as screening method, in order to identify relevant input parameters. In this sense, the values of the resulting EE should be handled with care. They only provide an indicator of relevance, not of the sensitivity (or even the comparative relevance). This should be taken into account when evaluating the resulting EEs.

A.       Elementary Effects at its fundamental level can only be considered a screening method. However, the introduction of the use of Sobol numbers and radial trajectories increases its efficacy as a method for estimating sensitivity, not just as a screening method.  Campolongo empirically demonstrated that the results obtained by EE can converge to a variance-based sensitivity index with increased number of Sobol points.  In this work, the authors increased the number of Sobol starting points until the EE-based sensitivity metrics had shown convergence.

Page 1: "sensitive" to "influential:

A.      The authors believe that "sensitive" is the proper term in this case.

Page 1: are you sure? to my knowledge, EE is not suitable for the quantification of uncertainty..
A.      See explanation above.

Page 2: towards
A.      Due to the placement of this comment in the document, the authors are unsure what it is referring to.

Page 2: not 100% sure since no native english speaker myself, but shouldnt it be the "sensitivity of an output towards an input" rather than the other way around? in the same way i would write here "impact of input on response,..."
A.      I have changed it to "sensitivity due to each input"

Page 2: so different number of seeds for each design point? what was the employed convergence threshold?
A.      The same number of seeds was used for each design point. To establish a convergence threshold, the authors visually examined the convergence to ensure that differences between the EE values across the design points were larger than differences between seeds.

Page 3: somewhat out of touch with the motivation of EE, which is screening.
A.      This comment is in relation to the fact that we had to first choose a set of parameters for the study, which involved down-selecting from a large list of possible ones.  If all parameters would be considered, the total number could reach into the thousands.  The approach used here was to choose those that spanned the characteristics being considered, and in a succinct manner.  For instance, the TurbSim model parameters were chosen as an effective representation of the parameters for wind-inflow characterization.

Page 3: this figure is very high level and does not contain a lot of information, but does take a lot of space. consider leaving out?
A.      The authors believe that this figure aids in the explanation on the setup and considered parameters.

Page 3: i think in sensitivity terminology this would be the dependent or output parameters. it might be valuable to stick to the same nomenclature in order to avoid confusion?
A.      The authors have changed the text to consistently use "QoI" throughout the text.

Page 3: vector sum based on the time series or on the ULS results of the time series?
A.      The vector sum was taken at each time step.

Page 4: in table 1
A.      This sentence is referring to the processes of used to assess the sensitivity of the input parameters, which is separate from the QoIs described in Table 1. This has been clarified in the text.

Page 4: not sure if this is correct ("common")

A.    The authors believe this is correct.

Page 4: including meta modeling
A.    We are unsure of the meaning of this comment.  The sentence is about meta modeling.

Page 4: dont understand this comparison. the choice of sensitivity analysis is more or less independent from the choice of sampling procedure..
A.    The discussion here is on how to improve the time efficiency of a sensitivity analysis. This can be done either through the mathematical approach for calculating the sensitivity, or through a down-selection of simulations based on the important ones, or through a reduced-order model.

Page 4: elementary effect
A.    I'm assuming that the reviewer would like us to spell out EE here?  The name has been introduced before, as well as the acronym, but we will go ahead and spell it out again here.

Page 5: R is 10 in this work => 10%?
how is this value selected?
A.    R is 30 in this work and was selected based on a convergence study that considered all QoIs.

Page 5: in line with proposed method from campolongo, no? should be sufficient to only highlight the differences from there
A.    Compolongo has looked at multiple approaches, so the authors thought it would be useful to provide a high-level summary of the approach being taken.

Page 5: reference?
A.    A reference has been added.

Page 6: clarify again that i is for the input variables
A.    The authors to not believe this is necessary, as the parameter is defined multiple times before this point.

Page 6: r is iterating from 1 to 10?
A.    30 starting points are used in this work. $r$ is not necessarily iterating from 1 to 30, but is instead being calculated separately at each starting point.

Page 7: difficult statement. EE already describes the sensitivity. Do you have an "exact" sensitivity in mind that is modeled with the EE?
A.    There is a sensitivity formula suggested by Campolongo, and introduced by Janson.  In the end, the formula is very similar to the mu* metric associated with Elementary Effects, and is therefore used directly as a sensitivity metric.

Page 7: can you explain your motivation of deviating from from the classical procedure? is this resulting from introducing new measures of EE which deviates from general methodology?

A.      The classical method has a pictorial representation of mean vs standard deviation of the EE values.  There is no definitive procedure for identifying or selecting the most significant parameters.  The metric devised here is focused only on mu* (mean) as this is the sensitivity indicator, whereas the standard deviation is an indicator of the influence of other parameters.

Page 7: is there a general procedure on how to assess these thresholds?
A.      The authors chose a value that was reasonable given the data.

Page 7: also an interesting paper from kelly and dimitrov using pce to establish sobol indices
A.      Thank you for the additional reference. However, since this was not used when establishing the method we did not add it as a reference in the paper.

Page 8: should be highlighted in the abstract that a kaimal spectrum was used
A.      A statement as been added to the abstract.

Page 11: you show results across all QoI, right?
A.      Correct. These results are presented in Figures 5-8 and Figures 15-18.

Page 24: this seems very large
A.      The level of yaw error was based on a reference by Quick, and the author's agreed that this level was feasible by consulting experts.

Page 27: should be mentioned in section on methodology or outlook as well
A.      The general methodology is summarized above.  There were issues encountered for this second case study, and we felt it was best to discuss the issues encountered within the individual study, since they did not occur for the other case study.

Page 27: this paragraph should be revised to increase readibility.
A.      The paper will go through a formal review process with an editor. This will be addressed at that time.

Page 36: the purpose of EE is more the determination of relevant input parameters, not the sensitivity assessment. This should not be the motivation of this study.
A.      Please see discussion above.

Page 36: is there any imporvement possible on the applied procedure in this work for other systems or do you propose that it can be used as a baseline method?
A.      Though minor adjustments may need to be made, the overall process is quite applicable to the related analyses.

---

## Author Comment (AC6) · 15 Apr 2019

**Authors' Note to the Associate Editor and Reviewers**

**Title: Sensitivity of Uncertainty in Wind Characteristics and Wind Turbine Properties on Wind Turbine Extreme and Fatigue Loads**

**Ref. No: wes-2019-2**

The authors thank the reviewers for their thorough assessment, comments, and insights. Revised text in the manuscript is marked in red.

Reviewer's comments are shown in blue. Authors' responses are shown in black.

1 General comments

1.1 Summary of key points

The paper is well written and its topic is relevant. The goals are clearly stated: sensitivity analyses for the NREL 5 MW turbine. They are ambitious because a large number of input and output variables are involved and a computationally demanding model (OpenFAST-based) is used. The choice to reduce the complexity of the analysis by using the relatively simple Elementary Effects approach is judicious. However, it seems to me that nevertheless, the problem is still too complex. To be able to tackle it, the authors work with a relatively small set of input vectors. This is the main weakness I find to be present in their analysis and they have not convincingly argued that the number of input vectors is sufficient.

A.       The authors parameterized inflow inputs based on the capabilities of TurbSim and parameterized the turbine inputs to represent the main physical effects where uncertainties were probable.

A further issue is their adaptation of the Elementary Effects approach in a way that is insufficiently justified. The current exposition leaves me doubting that it is really a consistent sensitivity analysis approach. This does not mean that all their conclusions are arbitrary. On the contrary, I would guess that many conclusions about sensitivities are correct due to their broadly consistent nature over the whole input space and remain unaffected by details of the sensitivity analysis. (This may mean that they would also appear in simpler analyses and could perhaps be obtained from expert elicitation.)

A.       Elementary Effects at its fundamental level can only be considered a screening method. However, the introduction of the use of Sobol numbers and radial trajectories increases its efficacy as a method for estimating sensitivity, not just as a screening method.  Campolongo empirically demonstrated that the results obtained by EE can converge to a variance-based sensitivity index with increased number of Sobol points.  In this work, the authors increased the number of Sobol starting points until the EE-based sensitivity metrics had shown convergence.

Despite my rather negative judgment about the method and assumptions, there are some gems in this paper. Notably, the authors' efforts in obtaining useful ranges for input variables have resulted in overview tables that are more broadly useful in their own right.

A.       Agreed.

1.2 Overview of review aspects
My judgments here are based on my current understanding of the work. Brief justifications are given, but detail and nuance for the negative comments are postponed to the 'Specific comments' section. They may change due to clarification by author comments.

1. Does the paper address relevant scientific questions within the scope of WES?
Yes. Knowledge about sensitivities is of great value to wind turbine design and selection.

2. Does the paper present novel concepts, ideas, tools, or data?
Some. Novel variants of Elementary Effects sensitivity analysis are presented. A lot of interesting, new simulation data was generated and used.

3. Is the paper of broad international interest?
Yes. The discussion is relevant for all locations and in various wind energy research subdomains.

4. Are clear objectives and/or hypotheses put forward?
Yes. To provide sensitivities of relevant output variables relative to coherent sets of input variables.

5. Are the scientific methods valid and clear outlined to be reproduced?
Validity may be tenuous and reproduction would be difficult. (i) I have my doubts that the novel variants of Elementary Effects sensitivity analysis is a proper sensitivity analysis.
A.      See above discussion.

(ii) The determination of input variables is not discussed in sufficient detail for them to be even approximatively recreated.
A.      All input parameter ranges are referenced where possible. The authors believe this to be sufficient.

6. Are analyses and assumptions valid?
One important one is not. I find it doubtful that the set of input vectors is large enough to warrant conclusions as concrete as the ones drawn, even more so given that a nontrivial number of them may not correspond to physical situations.
A.      The authors attempted to select turbine input parameters that are independent from each other. To minimize correlations to inflow input parameters, the results were binned by mean wind speed.

7. Are the presented results sufficient to support the interpretations and associated discussion?
No. This is a consequence of my evaluation of the two preceding points.
A.      The authors have addressed these points.

8. Is the discussion relevant and backed up?
Yes. Based on the results the authors present, the conclusions are reasonable.

9. Are accurate conclusions reached based on the presented results and discussion?

Ambiguous. Given the presented results and discussion, the conclusions are accurate, but I think that flaws in the analysis and assumptions cast doubt on that accuracy.

A.      These concerns have been addressed above.

10. Do the authors give proper credit to related and relevant work and clearly indicate their own original contribution?
Yes. According to my knowledge they certainly do. And notably, they make very good use of information in the literature for ranges of input variable values.

11. Does the title clearly reflect the contents of the paper and is it informative?
It can be improved. Currently, the title implies a scope that is larger than in actuality and is a bit long and complex. Suggestion: "Elementary effects sensitivity
analyses of the NREL 5 MW turbine"

A.      The title has been modified.

12. Does the abstract provide a concise and complete summary, including quantitative results?
Yes, but. (i) Quantitative results are not given, but neither are they appropriate. (ii) The future applications listed are not sufficiently discussed in the paper to be included.

A.      The future applications provide the justification for doing the research in the paper and are therefore included in the abstract.

13. Is the overall presentation well structured?
Yes.

14. Is the paper written concisely and to the point?
Yes. There is a bit of repetition in the presentation of the second case study, but this redundance may actually make the paper easier to read.

15. Is the language fluent, precise, and grammatically correct?
Yes.

16. Are the figures and tables useful and all necessary?
Not all. I found Figures 5–8, 10–11, and 15–22 of limited usefulness; the information should be more filtered and summarized to be useful. In contrast to these stand especially Figures 9 and 23.

A.      Some figures have been moved into an appendix.

17. Are mathematical formulae, symbols, abbreviations, and units correctly defined and used according to the author guidelines?
Not all or in all aspects. Standards about notation of variables and constants are not followed and a mixture of fonts is distractingly used for mathematical notation.

A.      The authors have attempted to improve on this. Additionally, the paper will go through a formal review process with an editor. Any remaining issues will be addressed at that time.

18. Should any parts of the paper (text, formulae, figures, tables) be clarified, reduced, combined, or eliminated?
Yes. For example, Figures 5–8, 10–11, and 15–22 as mentioned above.

A.      Some plots have been moved to the appendix.

19. Are the number and quality of references appropriate?
Yes.

20. Is the amount and quality of supplementary material appropriate and of added
value?
No supplementary material has been provided. It would have been useful if the simulation data
(inputs, outputs) were made avaiable, but it is of course the prerogative of the authors not to do
so.
A.      The authors believe that this is impractical, especially for an already lengthy paper.

2 Specific comments
2.1 Your modified EE formulae
There is insufficient justification of your modified formulae. You indicate why you add Yob in
Eq. (3) and use a dimensional version, but just mentioning it is insufficient as an explanation. To
me, adding a constant term to a set of sensitivities will substantially distort the information
present in the quantity; it is not a sensitivity anymore. I can sense a reason for making it
dimensional, but I can think of other reasons why this is a bad idea; you should pre-emptively
remove such doubts. Currently, I am not convinced at all that what you call a sensitivity here can
be interpreted and used as such.
A.       The inclusion of Yob was necessary to properly compare ultimate load consistently
across the bins because it is only the maximum ultimate load that matters.

In Eq. (5), you multiply by a probability and again indicate why, but do not explain it at all.
Here, I can guess at the reason.
Part of my reticence here is due to the fact that I am skeptical of your approach to the
identification of most sensitive inputs; this is discussed next. As your modifications here are, I
assume, related to your non-standard approach, it may be good to explain them concurrently in
the text.
A.      The scaling by the probability was necessary to properly compare fatigue loads
consistently across the bins because it is the cumulative effect of all bins that matters for fatigue.

2.2 Your approach to the identification of most sensitive inputs
First of all, by looking at plain means, you implicitly assume that your samples are uniformly
distributed over the input space. Even if you cannot justify this, you should at least discuss the
implications. Related to this, you apparently do not calculate the expectation over wind speeds.
Of course, for fatigue loads this is actually what happens because you have included the
probability in the EE value. For ultimate loads, it does not, where I see no reason why you
should throw away the probability information that you do have here and replace it with a
uniform distribution (implicit by taking the plain mean, as said before).
A.      The authors looked at the convergence of the number of starting points. Starting points
were added until adding points no longer changed the mean of the EE values. Therefore, while
there may be improbable events included in the parameter space, they do not affect the overall
conclusions.

Second, you say that you do not use the standard approach, but do not really justify that. You refer to an appendix and there is information about that there, but take into account that appendices are meant for skippable material. When reading your approach and the standard one as sketched in the appendix, the latter was more appealing to me. The reason is that there the rankings are primarily based on the means.

A.    The authors found that the exceedance approach was more justifiable and intuitive for the purpose of this research. The standard approach results was included in the appendix for potential readers who are interested in such results, but were not found to be the most beneficial presentation of this research.

You base your rankings essentially on tail behavior of the EE value distribution. Tail behavior says very little about what happens in the bulk of the distribution. It may well be that a distribution with a low mean but a fat tail generates more instances above your somewhat arbitrarily defined threshold than a distribution with a high mean but a skinny tail. Based on this reasoning, I fear that your approach may unduly rank higher inputs that generate fat tails, based just on the tail and not on the mean. It is possible that the tail behavior is similar over the inputs and then your criterion will work, but you do not show or discuss that. Even in that case I see no reason not to focus more on the means. So, a good start to convincing me is to explain why a ranking of means is not appropriate here and how your criterion overcomes the problems apparently present in other approaches.

A.    The authors believe that the exceedance probability plots and histograms show such behavior and therefore allow for such concerns to be addressed in the analysis.

2.3 Your selection of input vectors
You use thirty input vectors times three—one for each wind speed bin—for both case studies. The quality of the selection of these points is essential, but the way in which you choose them is dealt with only in a single sentence where you claim that by using Sobol numbers—without providing details—you can ensure a wide sampling of the input hyperspace. Furthermore, you choose to ignore the dependencies between the input variables, but directly sample from the Cartesian product of the individual ranges you have defined, so non-physical input vectors could be included. Finally, your input spaces are very large, 18-dimensional and 40-dimensional respectively.

A.    As mentioned above, a convergence study on the number of starting points was performed and the authors attempted to minimize correlations between inputs.

In such high-dimensional spaces, even a few dependencies can cause the subset of physical vectors to be 'small' relative to the Cartesian product. So it may be difficult to actually land on a physical vector, when those dependencies are not taken into account. (I do not see how using Sobol numbers can help with this issue, as more even sampling cannot correct for ignored dependencies.) This may mean that I cannot exclude the possibility that the majority or even all of the input vectors you use is non-physical. The fact that 90 input vectors in such high-dimensional spaces is a very small sample, only makes this issue more problematic. (I understand that for computational time reasons you cannot increase this number by orders of magnitude.) Because a priori I must assume that non-physical input vectors may result in non-realistic output sensitivities, this issue undermines your results.

In the conclusion, you state that "The combinations of parameters in this study spanned the ranges of several different locations." Given the reasoning I developed above, you will understand that I am skeptical of this. But this can be tested: how representative are your input vectors of existing locations? This may provide an avenue to reduce my worries about your selection of input vectors.

A.      These concerns have been addressed above.

2.4 Your presentation of applications and future work
In the conclusions, you present a number of possible applications of your work and also ideas for future work. I feel that some—about error bars and insight—are described to briefly or too vague to really know whether indeed, your results provide a useful starting point.

A.      The authors believe it is important to focus on what was done in the paper. Since this resulted in a lengthy paper, it was decided to simply highlight potential future work to save space.

3 Technical corrections
3.1 General
• In mathematical notation, the following standard conventions are prescribed: variables are written in a cursive/italic/slanted font and constants are written in a roman/upright font. https://www.nist.gov/pml/nist-guide-si-chapter-10-more-printing-and-using-symbols-and-numbers-scientific-and-technical#102 Please do so throughout, as currently this is not adhered to and most every symbol is put in italic font, even word abbreviations (which are certainly constant).

A.      The paper will go through a formal review process with an editor. Any remaining issues will be addressed at that time.

• As per the SI standards, between a value and a unit there should always be a normal (unbreakable) space. So do not write '5-MW', but '5 MW'; there are other examples in the text where no space is present. Also, composed units should not be separated by a hyphen, use a centered dot instead; e.g., 'kN·m' instead of 'kN-m'.

A.      These corrections have been made throughout the paper.

• You abuse the same symbol for functions and variables: for example $Y$ and $Y_o$ instead of $f$ and $f_o$; avoid that, as it causes confusion.

A.      The paper will go through a formal review process with an editor. Any remaining issues will be addressed at that time.

• Your presentation of the EE approach is symbolically far more verbose than it needs to be. Suggestions:
–Make effective use of vectorial notation. For example, Eq. (2) could be written as
$$EE_i = f(\vec{U} + \Delta \vec{e}_i) - f(\vec{U})\Delta.$$
(Arrows instead of bold here only because of reviewing system limitations.)

A.      This has been somewhat addressed. The paper will go through a formal review process with an editor. Any remaining issues will be addressed at that time.

–Use consistent notation for elements (lower case) and sets (upper case). For example u∈U, y instead of Y.

A.     This has been somewhat addressed. The paper will go through a formal review process with an editor. Any remaining issues will be addressed at that time.

–Make subscripts more 'direct' by avoiding index layers and using variables directly. For example, v instead of both b and vb, u instead of both r and ur.

A.     This has been somewhat addressed. The paper will go through a formal review process with an editor. Any remaining issues will be addressed at that time.

–Put commas between subscripts and do not overburden subscript/superscript locations.

A.     This has been somewhat addressed. The paper will go through a formal review process with an editor. Any remaining issues will be addressed at that time.

–Predefine some recurring fragments, such as $\Delta vi = maxUvi - minUvi$ and $\delta vi = \Delta vi/10$.

A.     This has been somewhat addressed. The paper will go through a formal review process with an editor. Any remaining issues will be addressed at that time.

Combined, you could write Eq. (3), (4), and (5) as

$$EEvi,o(\sim uv) = \bar{y}vo + 10|fo(\sim uv \pm \delta vi \sim ei) - fo(\sim uv)|$$
$$\text{with } fo(\sim uv) = 1|S|\sum s \in Smaxt|fo,s(\sim uv,t)|,$$
$$EEvi,o(\sim uv) = 10|go(\sim uv \pm \delta vi \sim ei) - go(\sim uv)|p(v)$$
$$\text{with } go(\sim uv) = ...$$

A.     This has been somewhat addressed. The paper will go through a formal review process with an editor. Any remaining issues will be addressed at that time.

• You use 'parameters' where I would use 'variables', given that the focus is on varying those quantities. For example, in your setup, I would call wind speed a variable, as it parameterizes some sub-cases, but is not varied as, e.g, wind shear is.

A.     The authors chose to use "parameters" because that is the term used to define inputs in the aero-elastic software used in this research and makes the most sense in the context of this work.

• Throughout the paper, the formulation of sensitivities are according to my ear often reversed. In the paper, input variables are called '(most) sensitive', whereas I would apply such language to output variables only. For input variables, words like 'impactful' and 'influencing' come to mind, although these do not seem ideal. Consider changing the language, but for the input side feel free to find a better word than the ones I came up with.

A.     The authors prefer to keep the language as is, which has precedent in related publications.

• Do not break tables over multiple pages; if you must, repeat the header row.

A.     This is absolutely true. However, as formatting will change significantly with final publication, this has not been specifically addressed here.

3.2 Local
p.2, l.33 OpenFAST is software, not an approach.

A.       This has been changed in the text.

p.3, l.6 What does 'down-selected' mean?
A.       This has been changed to "selected".

p.3, l.13 You must say how this was assessed.
A.       This is thoroughly addressed in Section 3.

p.4, Table 1 This nice table can be improved by adding symbols and a clear indication of which variables are used for which case study (wind vs. turbine and ultimate vs. fatigue).
A.       The authors believe that the table is best presented as is. All QoIs are used for all the studies.

p.4, §3.1 Paragraph way too long; split in two or three.
A.       The paper will go through a formal review process with an editor. Any remaining issues will be addressed at that time.

p.5, l.20 Elaborate, the current explanation about the 'radial approach' is too limited.
A.       More explanation was provided in the paper.

p.5, l.22 Elaborate, the current explanation is too limited. How did you obtain the Sobol numbers and how did you use them.
A.       One can use Sobol sequences to fill up a space with "random" points. The nice thing about it is that the points distribute themselves fairly evenly and therefore sample the space uniformly (to a good extent) without having a pattern per se.

p.6, l.1 Say how and why it was modified or at least reference forward.
A.       A reference note was made in the paper.

p.6, l.8 'nondimensionalizing' to 'making dimensionless/adimensional'
A.       This change has been made.

p.6, l.9 'derivative' to 'finite difference'
A.       This change has been made.

p.6, l.10 'IEC turbine class I and category B' to 'IEC turbine class I and category B' (to avoid confusion with index sets I and B )
A.       This change has been made.

p.6, Eq. (3) and (5) $i_b \pm 1$ to $b_i \pm 1$ (but better still follow my general suggestions regarding math notation to avoid making such mistakes)
A.        The paper will go through a formal review process with an editor. Issues such as this will be addressed at that time.

p.6, Eq. (4) In principle, the LHS depends on the set of seeds

A.      The authors performed a convergence study on the number of seeds used in the study to ensure that the solution converged independent of the seeds.

p.6, l.20 Why choose the sign randomly? Does this matter?
A.      This allowed for more of the multi-dimensional input parameter space to be considered.

p.6, l.20 and elsewhere 'IEC-Class IB' to 'IEC Class IB'
A.      This change has been made throughout the text.

p.6, l.20 move 'The added term . . . ' forward in section.
A.      This section has been rewritten.

p.6, l.23 Define DEL (as you should all letter words); do not assume all your readers will be familiar with this.
A.      This term was previously defined on page 3, line 22.

p.7, l.6 Your notation includes indices that are in fact averaged away (r, i, b) for the mean EE value, but for the (sample) standard deviation, you include not even the one that isn't gone (o). Some consistent, standard notation would be appropriate here; consider something like the sample mean mo and sample standard deviation so, perhaps.
A.      The authors believe that once EE has been clearly defined, it is acceptable to simply state that the EE values have a mean and standard deviation.

p.7, l.8 The part about 'stratification' requires more explanation, certainly because it is not immediately clear in what way this differs between the two case studies. (Perhaps you can reference to a relevant earlier part of the text.)
A.      This has been clarified.

p.7, l.8–12 The way in which you decide to include or exclude some wind speed bins comes accross as somewhat arbitrary here. More explanation may be needed.
A.      This section has been clarified.

p.7, l.17 'speed bins' to 'speeds'
A.      The paper will go through a formal review process with an editor. Issues such as this will be addressed at that time.

p.7, l.24 'Holtstag' to 'Holtslag'
A.      This has been corrected.

p.7, l.35 'either better optimized or lower risk': optimized for what? lower risk of what? Avoid vagueness here, as concrete applications are a good justification for your work.
A.      By increasing accuracy, you are eliminating uncertainty in your analysis.  A system would be better optimized if, based on reducing uncertainty, you are able to narrow your design margin.  However, if your uncertainty was actually allowing you to have a design that had a higher probability of failure than expected, lowering this uncertainty would therefore reduce the risk of failure, as the system would be appropriately re-designed.

p.8, Table 2 use the same, correct mathematical symbol, e.g., 'u(σu)'.
A.      This nomenclature is consistent with the IEC wind turbine design standard.

p.8, l.16 Suggestion: for q, use the velocity component name directly, instead of an index.
A.      The paper will go through a formal review process with an editor. Issues such as this will be addressed at that time.

p.8, l.19–21 Giving the exact quantity definitions of the IEC standard here is superfluous, certainly because you do not use it.
A.      The authors believe these definitions are important in the context of this work.

p.8, l.22 'and random' to 'and to the random' (?)
A.      The authors believe the grammar is fine as is.

p.9, l.9–10 'term' to 'factor'
A.      This has been corrected.

p.9, l.11 'in (IEC, 2005)' to 'in the standard (IEC, 2005)' (text should make sense when reading aloud, skipping over citation parentheticals)
A.      This has been corrected.

p.9, l30 'B' to 'B'
A.      This has been corrected.

p.10, Table 3 header
• There is no need for parentheses around units.
A.      The authors disagree. This is standard practice.

•bu,bv,bw are not dimensionless; I suggest using 1/mm for convenient notation of the values.
A.      This has been corrected.

p.10, Table 3 footnote Elaborate a bit.
A.      At extremely negative values of the shear exponent, $\alpha$, tower blade strikes would often occur. To eliminate this issue, the minimum value of $\alpha$ was changed to -0.75 from -1.5.

p.10, l.7 'unphysical' to 'non-physical'
A.      This has been corrected.

p.11, Fig. 3–4 Suggestion: use a logarithmic axis for the counts, then zooming will not be necessary and more information should become visible.
A.      The authors originally used a logarithmic scale on the y-axis. However, it was found that the figures are easier to interpret without a logarithmic axis.

p.19, Tables 4–5 Make it explicit in the table whether the numbers given are percentages or counts.
A.      This has been clarified in Tables 4, 5, 12, and 13.

p.20, l.8–9 Provide more information about the expert(s) and the elicitation procedure used.
A.      When references were not available for specific parameters, experts in the area of study at the NWTC were sought out.

p.20, Table 6• Do not use the empty set symbol $\emptyset$ instead of the Greek letter phi $\varphi$.
A.      This has been corrected.

•$B_{M,imb}$ to $B_{M,imb}$ (just an example of the type of math font use corrections that you should implement; note that the font of the subscripts, which here refer to words, are upright)
A.      These corrections have been made throughout the paper.

p.21–22, list around page break What are the parenthetical, bold math symbols for?
A.      They are identifying the symbols that are associated with the described parameters.

p.22, Figure 12 Avoid putting figures in the middle of paragraphs.
A.      This is something that will be changed with final formatting. As such, it is not addressed here.

p.23, l.18–19 Avoid line breaking tuples.
A.      Again, this will be changed in final formatting and is not addressed here.

p.23, l.19–20 Use paragraph breaks as defined by the style.
A.      Again, this will be changed in final formatting and is not addressed here.

p.24, Figure 13 •Are curves for $C_{d,orig}$ and $C_{d,-10\%}$ missing here, or do they just overlap?
A.      The lines overlap. This is explained in the text.

• This Figure is not referenced in the text, I think. (All figures and tables should be.)
A.      This reference was accidently deleted during formatting and has been added back in on page 23, line 17.

p.25, l.8–23 This paragraph is too long; split.
A.      While the paragraph might be lengthy, the authors believe that all of this information belongs in one paragraph.

p.25, l.17 'relevant' to 'relative' (?)
A.      This change has been made.

p.27, l.3–17 This paragraph is too long; split.
A.      While the paragraph might be lengthy, the authors believe that all of this information belongs in one paragraph.

A.      This has been corrected.

A.      Plots of σ vs mu are standard practice in EE literature. Since these plots were included to show the results in the standard way, they are kept as is.